# DAME: A Distillation Based Approach For Model-agnostic Local Explainability

## Abstract

The frameworks for explaining the functional space learned by deep neural networks, also known as eXplainable AI (XAI) models, are majorly based on the notion of the *locality*. Most of the approaches for local model-agnostic explainability employ linear models. Driven by the fact that a linear model is inherently interpretable (linear coefficients being the explanation), they are used to approximate the non-linear function locally. In this paper, we argue that local linear approximation is inapt as the black boxes under investigation are often highly non linear. We present a novel perturbation-based approach for local explainability, called the Distillation Approach for Model-agnostic Explainability (DAME). It separates out the two tasks- local approximation and generating explanation, and successfully attempts generating explanations by operating on high dimensional input space. The DAME framework is a *learnable*, saliency-based explainability model, which is post-hoc, model-agnostic, and requires only query access to the black box. Extensive evaluations including quantitative, qualitative and subjective measures, presented on diverse object and sound classification tasks, demonstrate that the DAME approach provides improved explanation compared to other XAI methods.

## 1 Introduction

Deep neural networks, shown to achieve human-level or even super-human performance for various tasks, have become the defacto framework for a variety of domains such as computer vision, natural language processing, and speech processing. However, the model's high accuracy can often be overestimated, as shown by Patel et al. (2008). In a related work, Kaufman et al. (2012) highlighted that the model outputs may not account for data leakage. Doshi-Velez et al. (2017) identified domains with a strong demand for accountability before model deployment, such as finance, healthcare, law, autonomous driving, and agriculture. Explainable artificial intelligence (XAI) aims to justify decisions made by deep neural networks, which is crucial for applications in biomedical, defense, autonomous driving, legal, and policy-making domains (Lipton, 2018; Holzinger et al., 2017; Caruana et al., 2015). Moreover, XAI methods help identify spurious correlations (Liao et al., 2020; Chouldechova & Roth, 2020), erroneous predictions (Gururangan et al., 2018), and reduce bias Ras et al. (2022); Wang & Rudin (2015).

The paradigm of *post-hoc* explainability methods is designed to operate within the context where the black box network, that requires an explanation, has already undergone training, with no possibility of retraining or modifying the black box model. Most of the post-hoc models are *local* methods since they provide explanations for single input-output pairs. Post-hoc methods can be categorized according to their applicability to different black box networks: (a) **model-agnostic** approaches, which are independent of the black box architecture (for example, Smilkov et al. (2017a), Sundararajan et al. (2017), and Ribeiro et al. (2016)), and (b) **model-specific** approaches, which are tailored to specific types of black box models, such as CNN-based models (for example, Zhou et al. (2016), and Chattopadhay et al. (2018)). Additionally, post-hoc methods can be classified based on the level of access to black box to generate the explanations. Broadly, there are two categories - (a) **Gradient-based methods**, which require access to gradient computations (Smilkov et al. (2017a), Sundararajan et al. (2017), Chattopadhay et al. Chattopadhay et al. (2018)), while (b) **perturbation-based methods** use local perturbations to generate explanations ( Zeiler & Fergus (2014), Ribeiro et al. (2016), and Petsiuk et al. (2018)) through *query access* (input-output access). Perturbation-based methods are more versatile and suitable for a wide range of applications The development of such

post-hoc methods is particularly critical as a number of large models are released with only query access (e.g. ChatGPT cha (2022)).

The perturbation requirements become significantly more challenging as the dimensionality of the data increases. For data modalities like images and audio, these requirements become extremely high (e.g., $5.0 \times 10^4$ for a $224 \times 224$ image). Existing post-hoc perturbation based approaches, such as locally invariant model-agnostic explanations (LIME) (Ribeiro et al., 2016), simplify the problem by significantly reducing the dimensionality. However, applying this technique to such disentangled data manifold raises concerns regarding its reliability. For non-linear networks, the approximation error arising due to linear approximation can be significantly high and we show in Appendix A.1 that they becomes ineffective with increasingly non-linearity of black box. Other prominent model-agnostic perturbation based approaches like RISE (Petsiuk et al., 2018) involves random masking involving no correlation or context information about the input features resulting in instability issues (Sattarzadeh et al., 2021).

In this work, we propose a novel distillation based *learnable* explainability model called DAME which is model-agnostic, gradient-free, and requires only query access to the black box.

- DAME attempts to mitigate the linear approximation error because even a mild non-linearity of the approximator can significantly improve the approximation quality as shown Figure 1.

- It separates out the two tasks - the local approximation (student network, $R$) and finding explanation (mask-generation network, $G$). Both $G$ and $R$ are learnable. $G$ is trained to select only the necessary and sufficient information of the image required by black-box for making it's predictions. A mildly non-linear $R$ is trained to distil the black box locally. The training is performed with fidelity maximization objective, which involved minimizing the mean square error (MSE) loss between black box and DAME predictions.

- DAME operates on the image space directly (same as black box), and hence, it can generate precise explanations.

We provide comprehensive qualitative and quantitative evaluation of DAME, comparing it with existing post-hoc methods. Further, we explore diverse tasks in image and audio domains for the empirical evaluations.

## 2 BACKGROUND

### 2.1 GRADIENT-BASED METHODS

The gradient-based approaches are based on the principle that the most discriminative regions of an input image should be associated with the large gradients. A few gradient-based methods are model-agnostic, for example, vanilla gradients (Simonyan et al., 2014), integrated gradients (Sundararajan et al., 2017), SmoothGrad (Smilkov et al., 2017a), and input multiplied with gradients (Shrikumar et al., 2017). Moreover, there are methods that are specifically designed for particular black box's architectures (e.g., class activation maps Zhou et al. (2016), GradCAM and variants Selvaraju et al. (2017)). Naturally, they are somewhat restrictive. Gradient-based approaches encounter several challenges, including generating noisy saliency maps, inaccuracies in the explanations (Adebayo et al., 2018; Mahendran & Vedaldi, 2016), and the issue of gradient saturation within the network (Shrikumar et al., 2017; Smilkov et al., 2017b).

### 2.2 PERTURBATION-BASED (GRADIENT-FREE) METHODS

Perturbation-based methods belong to the category of explainability techniques that find local explanations by evaluating the black box's response to small perturbations applied to the input. The two most prominent methods are LIME (Ribeiro et al., 2016) and RISE (Petsiuk et al., 2018).

**Locally Invariant Model-agnostic Explanations (LIME)**: LIME (Ribeiro et al., 2016) exploits the facts that a non-linear function can be approximated as a linear function within a small locality, and they are inherently interpretable. It learns a linear mapping between the perturbed input samples and the query output from the black box. However, the linear approximation can be erroneous in mimicking a non-linear black box as shown in Figure 1.

Due to the high dimensional inputs like images, LIME uses image segments (also called super-pixels) as features to avoid the computational burden. The binary information about which segments are masked on/off and associated black box responses are used to train the linear model. The linear coefficients represent the importance scores of different segments. The top-$k$ segments are chosen as explanations. A detailed description on working of LIME is provided in Appendix A.2. Narodytska et al. (2019) argue that LIME operates on a subspace of the input and hence, the explanations may not be accurate. Moreover, when the black-box model has a higher degree of non-linearity, a local linear approximation may not be optimal, as shown in Figure 1.

**Randomized Input Sampling for Explanation (RISE)**: RISE (Petsiuk et al., 2018) is a technique based on masking out different regions of the local input and using the black box response to find salient regions. The linear combination of the mask, weighted by the target-class probabilities, gives the saliency explanation generated by RISE. The approach uses randomly masked versions of the input image to generate multiple samples which leads to instability issues (e.g. Sattarzadeh et al. (2021)). Further details on the working of RISE is described in Appendix A.3.

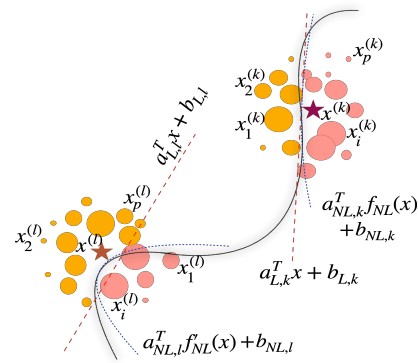

Figure 1: A local linear approximation, $\{a_{L,k}, b_{L,k}\}$ (red dashed line) of a non-linear decision boundary (gray solid line) is obtained using perturbations $x_i^{(k)}$'s around a local input $x^{(k)}$. The goodness of linear assumption looses strength where the decision boundary becomes less smooth (e.g. at $x^{(l)}$). An importance computed with mild non-linearity $f_{NL,l}$ for such point may lead to a more accurate approximation (blue dashed curves).

## 3 LOCAL LINEAR APPROXIMATORS ARE NOT GOOD ENOUGH

We further present a simple experiment ($E$) to exemplify the large approximation error that can be induced by local linear approximators, as discussed in Figure 1. We synthesize a dataset, $S = \{X, Y\}$, $X : \{x_1, x_2, ..., x_N\}$, $Y : \{y_1, y_2, ..., y_N\}$ for a supervised binary classification task, where $x_i \in \mathbb{R}^2$ is drawn from two different class-conditional distributions for the first dimension. The second dimension of the data points are drawn from the same distribution, in order to localize the discrimination only on the first dimension of the data. Here, $y_i \in \{0, 1\}$ indicating a binary classification setting. We train increasingly non-linear black box models (deep feed-forward networks) on this dataset. We then introduce a notion of eXplanation Error ($XE$) and show that with an increasingly non-linear black box, the explainability error with a linear approximation methods like LIME increases drastically. This set up is detailed in Appendix A.1.

## 4 DAME MODEL

### 4.1 PROBLEM FORMULATION

For post-hoc explainability, let the trained black box network, parameterized by $\theta$, be denoted as, $f_{\text{BB}}(\cdot; \theta)$. Let $X = \{x^{(1)}, x^{(2)}, \ldots, x^{(N)}\}$ and $Y = \{y^{(1)}, y^{(2)}, \ldots, y^{(N)}\}$ be the input and the corresponding soft-max outputs of the black box network, respectively. We formulate our method in a generic classification setting, where $f_{\text{BB}}(\cdot; \theta)$ is a classifier, $|f_{\text{BB}}(x^{(k)}; \theta)| = C$, and $C$ represents the number of classes. Let, $y^{(k)} = f_{\text{BB}}(x^{(k)}; \theta)$ and $y_T^{(k)} = [y^{(k)}]_T$ represent the target-class index $T$, for which the explanation is sought. Given $(x^{(k)}, y^{(k)})$ and target-class $T$, the local explainability problem is to obtain a saliency $E_T^{(k)}(x^{(k)}, y^{(k)}; f_{\text{BB}}(\cdot; \theta))$ for the input $x^{(k)}$.

### 4.2 PERTURBATION-BASED NEIGHBOURHOOD SAMPLING

We perturb the input, $x^{(k)}$ to generate $p$ local samples, $Q_{k,p} = \{x_1^{(k)}, ..., x_p^{(k)}\}$. We consider $x^{(k)}$ to be 2D (e.g. a gray-scale image) or 3D (e.g., a color image). We adopt the perturbation strategies that

embed contextual meaning. Particularly, we use image segmentation based perturbation (similar to LIME (Ribeiro et al., 2016)). Let the set of $Z$ segments extracted from $x^{(k)}$ be represented by,

$$S_{x^{(k)}} = \{\zeta_i^k\}, i \in \{1, 2, ..., Z\}; \zeta_i^k \subseteq x^{(k)} \forall i; \bigcup_{i=1}^{Z} \zeta_i^k = x^{(k)}; \zeta_i^k \cap \zeta_j^k = \phi, i \neq j \quad (1)$$

The $\zeta_i^k$'s are non-overlapping super-pixels (subset of pixels in $x^{(k)}$). The $\zeta_i^k$'s are randomly masked to generate the local neighbourhoods $x_i^{(k)} (\in Q_{k,p})$. The indices of segments being masked out to generate the $i^{\text{th}}$ neighbourhood are denoted as $I_i^k \subseteq \{1, 2, ..., Z\}$, $i \in \{1, 2, ..., p\}$. Let $M_{x^{(k)}} \in \mathbb{Z}^{p \times Z}$ be a matrix, representing binary information about segments being present/absent, given by,

$$[M_{x^{(k)}}]_{i,j} = \mathbb{1}_{\{j \notin I_i^k\}}, i \in \{1, 2, ..., p\}, j \in \{1, 2, ..., Z\} \quad (2)$$

Note that, the $i^{\text{th}}$ row of $M_{x^{(k)}}$ represents the $i^{\text{th}}$ local neighbor, $x_i^{(k)}$. Existing local approximators, like LIME (Ribeiro et al., 2016), use $M_{x^{(k)}}$ to learn the local explanation. Evidently, the domain of $M_{x^{(k)}}$, $\mathcal{D}(M_{x^{(k)}}) \in \{0, 1\}^{p \times Z}$, is a binary disentangled data manifold containing only masking index ($I_i^k$) information. In the DAME approach, we generate neighborhood samples, $x_i^{(k)}$ in the input space itself. We then utilize these samples to find explanations.

## 4.3 THE HADAMARD PRODUCT NOTION OF "EXPLANATION"

Let the true feature importance weights be denoted as $E_T^{(k)}(x^{(k)}, y^{(k)}; f_{\text{BB}}(\cdot; \theta)) = \{i_m^k\}_{m=1}^{D} \in [0, 1]$, $x^{(k)} \in \mathbb{R}^D$. Based on the notion of explainablity, attenuating/masking input features of high class-importance leads to a significant drop in model's posterior probability estimate of the target class and vice versa. The optimal set of explanations $\{i_m^k\}_{m=1}^{D}$ is the best attenuating function on dimensions of $x^{(k)} \in \mathbb{R}^D$ with the least change on the posterior probability estimate of the target class. Hence, under the assumption, $i_m \neq i_n, m \neq n$, a true explanation satisfies,

$$[f_{BB}(x^{(k)}; \theta)]_T - [f_{BB}(E_T^{(k)} \odot x^{(k)}; \theta)]_T \leq [f_{BB}(x^{(k)}; \theta)]_T - [f_{BB}(P(E_T^{(k)}) \odot x^{(k)}; \theta)]_T \quad (3)$$

where $E_T^{(k)}$ represents $E_T^{(k)}(x^{(k)}, y^{(k)}; f_{\text{BB}}(\cdot; \theta))$, and $P(\cdot)$ is a random permutation. Conversely,

$$E_T^{(k)} = \arg\min_P \left( [f_{BB}(x^{(k)}; \theta)]_T - [f_{BB}(P(E_T^{(k)}) \odot x^{(k)}; \theta)]_T \right)^2 \quad (4)$$

$$E_T^{(k)} = \arg\min_m \left( [f_{BB}(x^{(k)}; \theta)]_T - [f_{BB}(m \odot x^{(k)}; \theta)]_T \right)^2 \quad (5)$$

Consequently, *learning* an Hadamard product attenuation starting from a random set of weights while minimizing the model confidence drop for target-class, leads to the explanation $E_T^{(k)}$. Note that, the optimization with respect to Eq. (5) is ill-posed which results in trivial identity learning $E_T^{(k)} = I_{D \times 1}$. To avoid this, a penalty constraint on the magnitude of $E_T^{(k)}$ is provided as an $L_1$-loss given in Eq. (8).

## 4.4 BLACK BOX DISTILLATION

Building a learnable explanation framework based on Eq. (5) is not possible in a post-hoc gradient-free framework. It needs the gradients to flow through the black box. Thus, we propose to distill the black box model in the locality of the input sample in order to generate the explanations. In particular, using the perturbations $x_i^{(k)}$ and the associated black box response, we distill the black-box in a teacher-student manner using the mean-square error (MSE) loss.

## 4.5 MODEL ARCHITECTURE

To compute the saliency explanation, we have two tasks in hand: learning the black box model distillation and optimizing an attenuating function to obtain the explanation. We combine the two tasks using the perturbations $x_i^{(k)}$ for both the distillation and explanation learning. The learnable explainabilty framework of DAME is shown in Appendix Figure 9. First, the perturbed samples,

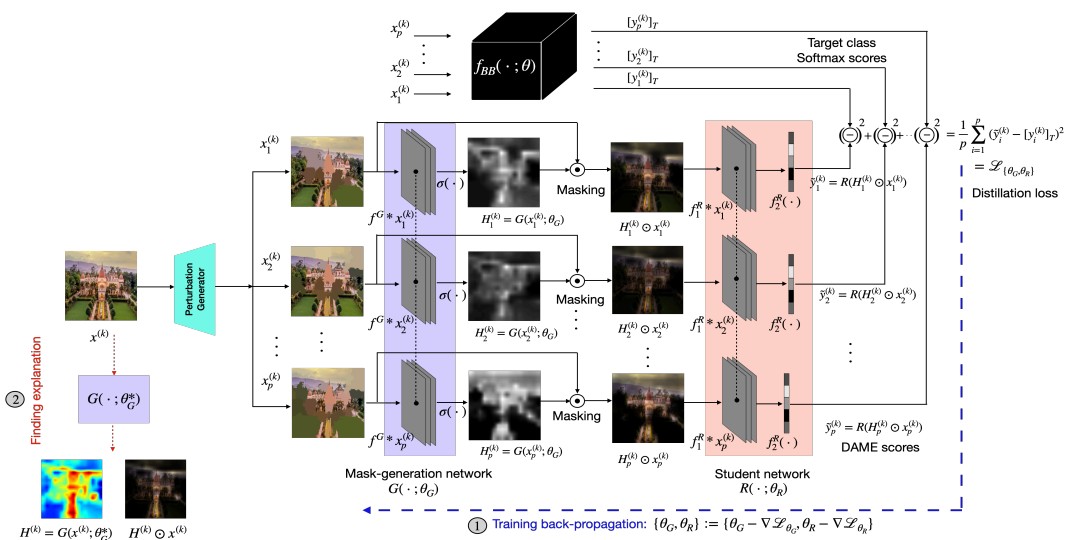

Figure 2: Architecture and training paradigm of DAME.

$x_i^{(k)}$ are passed through a mask generation network, $G(\cdot; \theta_G)$, that generates saliency maps, $H_i^{(k)}$. The Hadamard product of the saliency map, $H_i^{(k)}$, with $x_i^{(k)}$ is finally passed to the student network, $R(\cdot; \theta_R)$. The student network tries to approximate the black box target class predictions. The parameters $\{\theta_G, \theta_R\}$ are trainable. As described in Sec. 4.3 and Sec. 4.4, the distillation and mask generation tasks can be trained with the same objective of minimizing the MSE between black box predictions, $[y_i^{(k)}]_T$ and student network outputs, $\tilde{y}_i^{(k)}$.

**Mask Generation Network**: Let, $H^{(k)} \in \mathbb{R}^{3 \times U \times V}$, be the learned saliency map. The value of $H_{ij}^{(k)}$ denotes the importance of $(i, j)^{\text{th}}$ dimension in $x^{(k)}$. We learn $H_i^{(k)}$'s directly from $x_i^{(k)}$'s by passing them through a simple CNN network, having 2 convolutional layers. As the training progresses, the model converges to generate the saliency based explanations of the input. We call it mask generation network, $G(\cdot; \theta_G)$ (Figure 2). In our experiments, the 3D-saliency maps (for 3 color channels), $H_i^{(k)}$, are obtained from the mask generation network.

$$H_i^{(k)} = G(x_i^{(k)}; \theta_G) = \sigma(f^G(\theta_G) * x_i^{(k)}), \ \ i \in \{1, 2, ..., p\} \quad (6)$$

where, $f^G(\theta_G)$ represents the CNN layers and $\sigma$ represents sigmoidal function.

---

**Algorithm 1** Saliency generation algorithm

**Inputs:** $f_{BB}(\cdot; \theta)$, $x^{(k)}$, $T$, **Output:** $H^{(k)}$
**Training:** Init.: $n_e \leftarrow$ no. of epochs;
$\theta_G \leftarrow$ random; $\theta_R \leftarrow$ random;
$p \leftarrow$ no. of perturbations; $e \leftarrow 0$
Compute $\{x_i^{(k)}\}_{i=1}^p$ and $y_i^{(k)} = f_{BB}(x_i^{(k)}; \theta)$
**while** $e \le n_e$ **do**

   Compute $H_i^{(k)} = G(x_i^{(k)}; \theta_G)$ using Eq. 6;
   Compute $\tilde{y}_i^{(k)} = R(H_i^{(k)} \odot x_i^{(k)}; \theta_R)$ using Eq. (7);
   Compute $\mathcal{L}^{(k)}$ using eqn. 8;
   $\mathcal{L}^{(k)} \leftarrow \mathcal{L}^{(k)} - \eta \nabla \mathcal{L}^{(k)}$;
   $e = e + 1$;

**end**
**Inference:** $H^{(k)} \leftarrow G(x^{(k)}; \theta_G = \theta_G^*)$; discard $R$;

---

**Mapping Network**: The Hadamard product of $H_i^{(k)}$ and $x_i^{(k)}$ selects the important features in $x_i^{(k)}$ to obtain the saliency map. The non-linear mapping network is also a simple model containing 2 CNN layers followed by a fully connected layer. The output of the model denoted as $\tilde{y}_i^{(k)} \in \mathbb{R}$.

$$\begin{aligned} \tilde{y}_i^{(k)} &= R(H_i^{(k)} \odot x_i^{(k)}; \theta_R) \\ &= f_2^R(f_1^R * (H_i^{(k)} \odot x_i^{(k)})) \end{aligned} \quad (7)$$

where $f_1^R(\cdot)$, $f_2^R(\cdot)$ represent the CNN layers and the fully connected layer respectively.

## 4.6 TRAINING AND LOSS FUNCTION

We train the mask generation and student networks jointly using the input-output pairs $x_i^{(k)}$ and $[y_i^{(k)}]_T$, $k \in \{1, 2, ..., p\}$. The loss function contains mean squared error and the $L_1$ loss on the generated explanation (to avoid identity mapping). Moreover, we explore an additional KL-divergence loss to ensure that the score distributions from the black box and DAME over the batch of samples are similar. The overall distillation loss function is,

$$\underbrace{\mathcal{L}^{(k)}\left(\{x_i^{(k)}\}_{i=1}^p, \{y_i^{(k)}\}_{i=1}^p; \theta, \theta_G^*, \theta_R^*\right)}_{\text{distillation-loss}} = \min_{\theta_G, \theta_R} \beta \underbrace{\sum_{i=1}^p \left([y_i^{(k)}]_T - \tilde{y}_i^{(k)}\right)^2}_{\text{distillation+explnation}} + \lambda \underbrace{\sum_{i=1}^p ||H_i^{(k)}||_{L1}}_{\text{avoid identity learning}} +$$

$$\underbrace{\mu KL\left(p_{\tilde{y}^{(k)}} \,||\, p_{[y^{(k)}]_T}\right)}_{\text{encourage similar score dist.}}; \; \lambda, \mu \geq 0$$

(8)

where $\beta_i = ||x^{(k)} - x_i^{(k)}||_{L2}$ are the relevance weights of $x_i^{(k)}$ as neighbors of $x^{(k)}$ based on their vicinity to $x^{(k)}$, $p(\cdot)$ denotes the distribution of the scores and $KL(.||.)$ is KL-divergence. Here, $\lambda$ and $\mu$ are hyper-parameters.

After the training, the mask generation network is forward passed with the input image, $x^{(k)}$ to generate the saliency maps. Given the black box $f_{\text{BB}}(\cdot; \theta)$, an input $x^{(k)}$ and the target class T, the steps to train the student network followed by obtaining the saliency explanation are described in Algorithm 1. The Appendix A.4 illustrates the learning using the three loss components, the progression of the saliency maps during the learning, while the hyper-parameter ablation studies are reported in Appendix A.5.

## 4.7 EVALUATION

We evaluate and compare the performance of DAME with several existing XAI methods belonging to three different categories. Both the qualitative and quantitative analyses are provided. In the qualitative evaluation, we illustrate the saliency map produced by the XAI models on a diverse set of local input images. However, the qualitative evaluation only provides insight into the performance of the models on a small set of examples, and may not be a good representative of their statistical performance. For the quantitative evaluation, we use the Intersection-over-Union (IoU) metric. For the ground truth for explanations, we use the human-annotated class segmentation masks (whenever they are available for the downstream task). To allow using ground truth segmentation masks for evaluation, we consider only samples which are correctly classified by black box (with posterior probability more than the mean value of all the target class predictions). The steps involved in computing the IoU metric are described in Appendix A.8.

## 5 RESULTS

### 5.1 TASK I - OBJECT CLASSIFICATION IN IMAGES

**Datasets**: We use Pascal VOC (Everingham et al., 2015) and ImageNet (Deng et al., 2009) datasets, for qualitative and quantitative analysis respectively, which contain real life images and often have multiple objects in the image with complex backgrounds. ImageNet dataset contains 1000 object classes and Pascal VOC has 20 object classes. Pascal VOC additionally contains human annotated object segmentation masks. Details about usage of the datasets are provided in the Appendix A.6.1.

**Black box classifiers**: We have used the pre-trained ResNet-101 model (He et al., 2016) and vision transformer (ViT) (Dosovitskiy et al., 2021) models as black box classifiers. The pre-trained models, without the target layer, are affixed with a softmax layer (of 20 output neurons), and the last layer is trained on the development set of Pascal-VOC set of 20 classes. The ResNet-101 model gave a test classification accuracy of 86.1% for these 20 classes, while the ViT model achieved an accuracy of 92.3%. More details are provided in Appendix A.6.1.

**Qualitative Evaluation**: We perform the qualitative analysis of the explanations provided by DAME on images on ImageNet classes. We use a held out set of images to optimize the hyper-parameters of

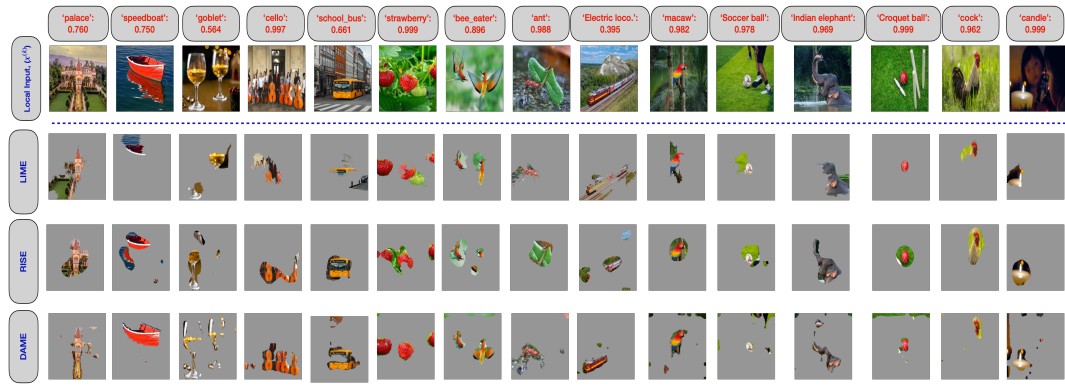

Figure 3: Qualitative assessment of different model agnostic gradient-free XAI methods, along with DAME, applied to the ResNet-101 black-box model. The target class for which explanation is sought and the corresponding black box confidence scores are mentioned. The explanations obtained are imposed on input images. Further extensive comparisons are shown in Appendix A.10 and A.9.

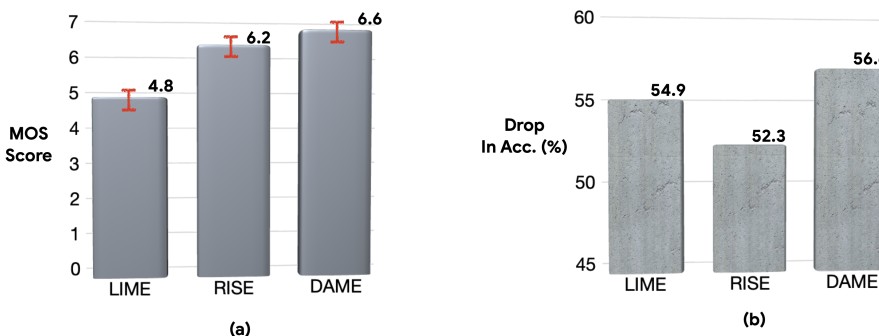

Figure 4: (a) Subjective evaluation with 35 subjects for scoring the XAI methods on 28 original images, (b) Counterfactual explanation evaluation – drop in accuracy on the target class for the black-box model by masking away salient regions identified by each of the XAI method.

the DAME before they are frozen for all the test images. Figure 3 shows the explanations provided by DAME and other standard methods. Also, same number of perturbations (3000 neighbourhood samples) are used for all perturbation based approaches. Followings are the key observations:

- On vast majority of the samples, DAME generates better explanations than linear approximation like LIME. The DAME, although operating on true image space of $\mathcal{O}(10^4)$ dimensions performs more consistently than many gradient based methods on some subset of the images (e.g. *speedboat*, *goblet*, *strawberry*, etc.).

- While operating on the image space itself, the DAME can generate very precise explanations (e.g. *electric-locomotive*). However, for certain images (e.g. *Indian-elephant* and *cock*), the DAME explanation does not capture the full object.

**Quantitative Analysis**: We evaluated the DAME approach using IoU metric, and compared it with the same existing approaches. Table 1 compares the performance of the proposed method with the 9 different baselines in terms of the IoU scores and for two different classifier models (Resnet-101 and ViT). For the Resnet-101 model, the gradient CAM based approaches give the best IoU values, although they require a gradient access to the model. For the ViT based classifier model, the DAME framework gives the best IoU results, among all the XAI methods compared.

### 5.1.1 SUBJECTIVE EVALUATION

We also performed a subjective evaluation of the 3 XAI methods. In this evaluation, we recruited human subjects and provided them with the original image, the label of the target class, as well as the saliency maps generated by the XAI methods. The subjects were asked to rate the quality of the

| Explainability Methods | | | Black box network | |
| | | | ResNet 101 | ViT-base-16 |
| Name | Gradient-free | Model-agnostic | Mean IoU (%) | Mean IoU (%) |
|---|---|---|---|---|
| Vanilla gradient | no | yes | 24.8 (8.1) | 19.6 (5.3) |
| Smooth-grad | no | yes | 34.5 (11.7) | 20.6 (6.4) |
| Integrated gradients | no | yes | 20.1 (8.5) | 19.3 (15.4) |
| Input x gradient | no | yes | 19.9 (8.7) | 19.2 (15.4) |
| GradCAM | no | no | 39.6 (13.9) | 16.7 (11.9) |
| Guided GradCAM | no | no | 19.0 (10.1) | 23.1 (15.4) |
| GradCAM++ | no | no | 39.7 (14.0) | 6.5 (7.7) |
| RISE | yes | yes | **33.7 (11.5)** | 31.1 (11.8) |
| LIME | yes | yes | 29.2 (10.9) | 26.8 (11.4) |
| **DAME** | **yes** | **yes** | 33.3 (12.1) | **31.4 (11.9)** |

Table 1: IoU (%) of 9 XAI methods and DAME on image classification task.

saliency map w.r.t. the class label on a scale of 1-10. There were 35 subjects who participated in this study. More details are given in Appendix A.7. The subjective results (average mean opinion score (MOS)) for the different XAI methods are provided in Figure 4 (a). In terms of the average MOS, the subjects preferred the DAME approach more than the other XAI methods (statistically significant with p-value $p \ll 0.05$, computed using a pair-wise t-test between DAME and RISE results).

### 5.1.2 FIDELITY BASED EVALUATION

The IoU metric compared the saliency map with the ground truth object segments to come up with a single measure of the overlap between the salient regions provided by the human annotation with the one provided by the XAI approaches. Another way of measuring the fidelity of the XAI methods is a feature removal based counterfactual evaluation Qi et al. (2019). In this setting, we mask the regions of the image that have been identified as salient by the XAI approaches. These masked images are input to the black-box classifier and the drop in model accuracy is measured. The intuition in this evaluation is that the XAI method, which captures the most pertinent regions for the black-box model in the original image, generates the largest drop

| Explainability Methods | Mean IoU (%) |
|---|---|
| Vanilla gradient | 15.5 (8.1) |
| Smooth-grad | 23.2 (6.7) |
| Integrated gradients | 20.1 (6.5) |
| Input x gradient | 19.1 (8.7) |
| GradCAM | 24.4 (11.9) |
| Guided GradCAM | 19.4 (4.6) |
| GradCAM++ | 25.7 (8.2) |
| RISE | 21.5 (4.5) |
| LIME | 17.0 (5.3) |
| **DAME** | **22.3 (4.2)** |

Table 2: IoU (%) of DAME and other methods on audio classification task. Among the gradient-free methods (last 3 rows), the DAME is seen to provide an improved IoU value.

in target class accuracy. Figure 4(b) illustrates the drop in prediction probability (%) of the target class for the black-box model using the salient regions identified by 3 XAI methods. As seen here, the DAME model generates the highest drop in counterfactual prediction accuracy.

### 5.2 TASK II - AUDIO EVENT DETECTION

**Dataset**: We use ESC-10 dataset Piczak (2015) for this task, which contains of diverse sound types, such as transient sounds, structured pattern noise, and harmonically rich sounds. In terms of size, the test data chosen for the analysis has 40 audio clips from each of the 10 classes. It is worth noting that, ESC dataset does not provide sound event annotation on the time axis. Further, the audio segments are well segmented in the original data. To facilitate statistical evaluation, we pad the audio with a random noise (between 1-5 sec. at 10 dB signal-to-noise ratio) at both ends of the audio clip. The clean portions of the modified audio recordings are considered as the "ground-truth" explanation, which is then compared with the model predictions.

| XAI Methods | Mean IoU (%) on the COSWARA dataset | | | | | |
|---|---|---|---|---|---|---|
| | Cough ($\uparrow$) | Throat clearing ($\downarrow$) | Noise ($\downarrow$) | Inhale ($\downarrow$) | Compress ($\downarrow$) | Silence ($\downarrow$) |
| RISE | 42.3 (4.5) | 26.5 (5.5) | 24.6 (4.9) | 13.5 (3.9) | **6.2** (2.5) | 6.1 (2.2) |
| LIME | 36.4 (6.3) | **16.8** (3.4) | 23.2 (5.8) | 18.3 (3.1) | 7.1 (2.3) | 5.1 (1.9) |
| **DAME** | **46.4** (5.2) | 25.1 (4.3) | **20.2** (4.1) | **12.3** (4.2) | 7.4 (3.5) | 5.9 (2.1) |

Table 3: IoU (%) of 3 XAI methods on the COVID-19 diagnostic task. The DAME approach is observed to associate the COVID-19 detection probabilities more to the cough region of the audio.

**Black box Classifier**: The ResNet-50 model, pre-trained on ImageNet dataset, is fine-tuned on the training subset of ESC dataset. The audio recordings are transformed into mel-spectrogram, which are served as the input for the model.

**Results**: The left part of Appendix Figure 12 shows the saliency explanation on the mel-spectrogram obtained using DAME for the black box prediction. We observe that most of the unique, repetitive patterns of the sound class are highlighted by the mask generation network. The saliency map is converted to a binary mask using a threshold. The right part of Appendix Figure 12 shows the mel-spectrogram of the modified audio clip. The modified audio clip only generated a minor change in the class probability of the true class ($0.68 \rightarrow 0.64$). As seen from the plot of $H^{(k)}$, DAME selectively picks up the patterns of the ground truth audio class and rejects the noisy regions. We statistically evaluate DAME on the ESC-10 dataset using IoU metric. The comparison between the proposed method, LIME and guided Grad-CAM, in terms of the mean and standard deviation IoU, is shown in Table 2, which reflects an improved performance of DAME over other baseline models.

## 5.3 Task III - COVID-19 Diagnosis from Cough Data

**Dataset**: For this task, we use the deep cough recordings from a publicly available Coswara dataset Sharma et al. (2020). Zhu et al. (2022) performed manual time-level annotations on the cough recordings, identifying different regions such as cough, throat clearing, noise, and silence.

**Black box Classifier**: As a black box classifier for this task, we use the bidirectional long short-term memory (BLTSM) network, as provided by Sharma et al. (2021). The classifier achieves a performance of $79.8\%$ area under the receiver operating characteristics curve for the COVID-19 diagnosis task. The model is trained on segments of a mel-spectrogram.

**Results**: We statistically evaluate the explanations provided by XAI approaches using manual annotations for $564$ cough recordings provided by Zhu et al. (2022). We compute the mean IoU value separately for different audio regions as given in the annotations. The IoU values of all the methods broadly agree (Table 3). As given by the mean IoU values of both classifiers, the unvoiced regions, namely the noise, silence, and inhale regions, are less salient compared to voiced regions (i.e, throat-clearing, cough). Moreover, the highest IoU over cough regions is provided by DAME.

## 6 Summary

In this paper, we propose a distillation based learnable approach for post-hoc explainability called Distillation Approach for Model-agnostic Explainability (DAME). It proposes a unique way to combine two non-linear networks to generate post-hoc explanations. Unlike existing perturbation approaches, it operates directly on the input image space instead of binary masks, yielding superior explanations based on qualitative and quantitative analysis. Although operating on such large dimensions ($\mathcal{O}^4$), as compared to other local approximators ($\mathcal{O}^1$), it requires a small number of perturbation samples, similar to the existing methods. By accessing true images while generating explanations, DAME generates very precise explanations, making it a promising direction for post-hoc explainability. The limitations include the need to train a non-linear model with hyper-parameters on the given input and with a $25\%$ more computational time per sample over the prior work of RISE. Other limitations of our approach may include sensitivity on the perturbation strategy being used to generate explanations, as seen for existing approaches. As future work, we intend to pursue the impact of the perturbations on the DAME performance across tasks.

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

## A   APPENDIX

### A.1   EXPLANATION ERROR FOR NON-LINEAR BLACK BOX

We further present a simple experiment ($E$) to exemplify the large approximation error that can be induced by local linear approximators, as discussed in Figure 1 in main draft.

#### A.1.1   DATASET

We synthesize a dataset, $S = \{X, Y\}$, $X : \{x_1, x_2, ..., x_N\}$, $Y : \{y_1, y_2, ..., y_N\}$ for a supervised binary classification task, where $x_i \in \mathbb{R}^2, y_i \in \{0, 1\}$. The data points $x_i$'s are synthesized so that their class-conditionals are sampled from different distributions for the first dimension ($D_0$ and $D_1$) while the second dimension for both the classes is sampled from same distribution ($D_2$). Particularly, $x_i[0]|_{y_i=0} \sim D_0$, $x_i[0]|_{y_i=1} \sim D_1$ and $x_i[1] \sim D_2 \forall y_i$, where $x_i[0]$ and $x_i[1]$ are first and second dimensions of $x_i$ respectively. The distributions are Gaussian mixture densities: $D_i(z) = \sum_{k=1}^{K} \alpha_k^i \mathcal{N}(z; \mu_k^i, \Sigma_k^i) \forall i$ with $\mu_k^i \neq \mu_k^j \forall i \neq j$ and $\alpha_k^i = \alpha_k^j, \Sigma_k^i = \Sigma_k^j \forall i, j$. $N$ is chosen as 10,000 and $K$ is varied to generate different versions of the dataset, denoted as $S(K)$.

#### A.1.2   CLASSIFIER

A fully connected neural network model (with 3 hidden layers and 50 neurons in each layer) is trained on the dataset for the binary classification task and then used as black-box for explainability analysis. Let the network trained on $S(K)$ be denoted as $M(K)$. The classification accuracy was kept constant with respect to different variants of the dataset, i.e. accuracy of $M(K)$ (which is trained on $S(K)$) is same for different values of $K$: $Accuracy(M(K); S(K)) = c, \forall K$.
Although the classifier architecture remains same thoughout, the non-linearity of it is controlled by varying $K$- that changes the dataset the classifier is trained on. We know, when class conditionals are unimodal Gaussian density, the Bayes optimum classifier is a linear classifier. Hence, for $K = 1$, $M(K)$, which is trained on $S(K)$, has linear decision boundary. For $K > 1$, $M(K)$ becomes increasingly non-linear because it is trained on class conditionals of multiple mixtures.

#### A.1.3   EXPLANATION USING LOCAL LINEAR APPROXIMATOR

Clearly from the dataset design, the first dimensions of $x_i$'s are the discriminatory features and the second dimensions are not. We take the data points from the test set that are correctly classified by the trained classifier $M(K)$ and use a local linear approximator to explain the classification. Let the set of correctly classified samples by the black-box $M(k)$ is $S_R(K) \subset S(K)$ with $|S_R(K)| = Accuracy(M(K); S(K)) \cdot |S(K)|$. An attribution based explanation $\{w_i^0 : x_i[0], w_i^1 : x_i[1]\}$ should put more weight on the first dimension than second i.e. $w_i^0 > w_i^1$. We define eXplanation Error ($XE$) as a function of $K$ ($0 \leq XE(K) \leq 1$) as,

$$XE(K) = \frac{|\{i : w_i^0 > w_i^1, (x_i, y_i) \in S_R(K)\}|}{|\{i : (x_i, y_i) \in S_R(K)\}|} \tag{9}$$

The experimental setup with dataset and classifier is shown in Figure 5. With increasing non-linearity in the black-box, $XE(K)$ is observed to significantly increase as shown in Figure 6.

### A.2   WORKING OF LIME

Below is a detailed description of generating explanations using LIME Ribeiro et al. (2016) as shown in Figure 7.

- The input images are of dimensionality $224 X 224$. As the images are directly used as input to the black box, a local approximation based technique needs to perform a very high dimensional ($\sim \mathcal{O}^4$) local approximation.

- Linear approximator like LIME drastically brings down the input dimensionality by performing image segmentation as shown in ① in Figure 7.

- The number of segments, $S$ are of $\mathcal{O}^1$ (30 for the image in Figure 7).

$X = \{x_1, x_2, \ldots, x_N\}, y \in \{0,1\}$

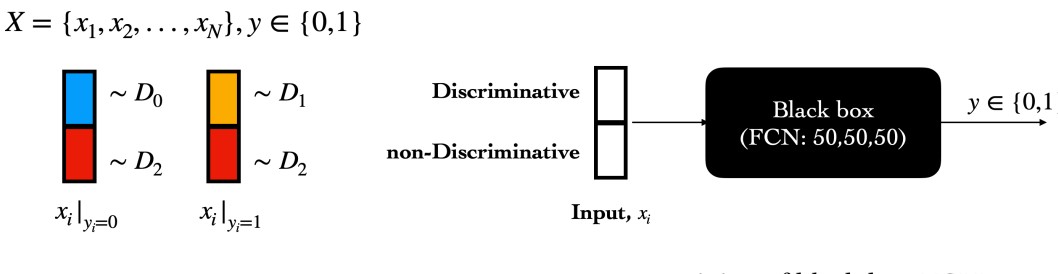

synthetic dataset                    training of black box FCN

Figure 5: Experiment (E) setup with the dataset and classifier details.

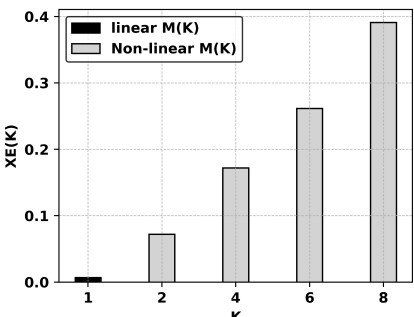

Figure 6: Through experiment (E), we show that with increasingly non-linear black box $M(K)$, the explanations generated by local linear approximators become significantly erroneous. Non-linearity of $M(K)$ increases with $K$. The explanation error, $XE(K)$ increases with $K$.

- It represents the original image as an $1XS$ binary array. All the entries being 1 denote that all the segments are unmasked.
- To generate $P$ random perturbations locally, it masks off some of the segments (randomly chosen). The corresponding binary representations are replacing the 1's by 0's for the masked-off segments. So, the local neighbourhood representations after masking becomes a $PXS$ matrix as shown in ① in Figure 7.
- The masked image representations are fed to the black box, and using the query access to it, the corresponding target class scores $t_1, t_2, ..., t_P$ are obtained as shown in ② in Figure 7.
- The local linear approximator, LIME fits a linear mapping from the binary matrix to the targets $t_1, t_2, ..., t_P$ as shown in ③ in Figure 7. This gives segment-wise importance scores as given by $w \in \mathbb{R}^{SX1}$ of the linear mapping $\{w, b\}$.
- The segment-wise importance scores can be used to choose top-$k$ important segments. They serve as the explanation as shown in ④ in Figure 7.
- LIME suffers from a few disadvantages: (a) The explanations provided are at segment-level, making the explanations imprecise, (b) It only uses disentangled binary masking information while generating explanations whereas the black box operates on the image space. It questions the reliability of the explanations, (c) The local linear assumption looses strength where the black box function locality becomes less smooth.

## A.3  Working of RISE

Below is a detailed description of generating explanations using RISE Petsiuk et al. (2018) as shown in Figure 8.

- RISE is based on weighting masks based on the corresponding model confidence drops.

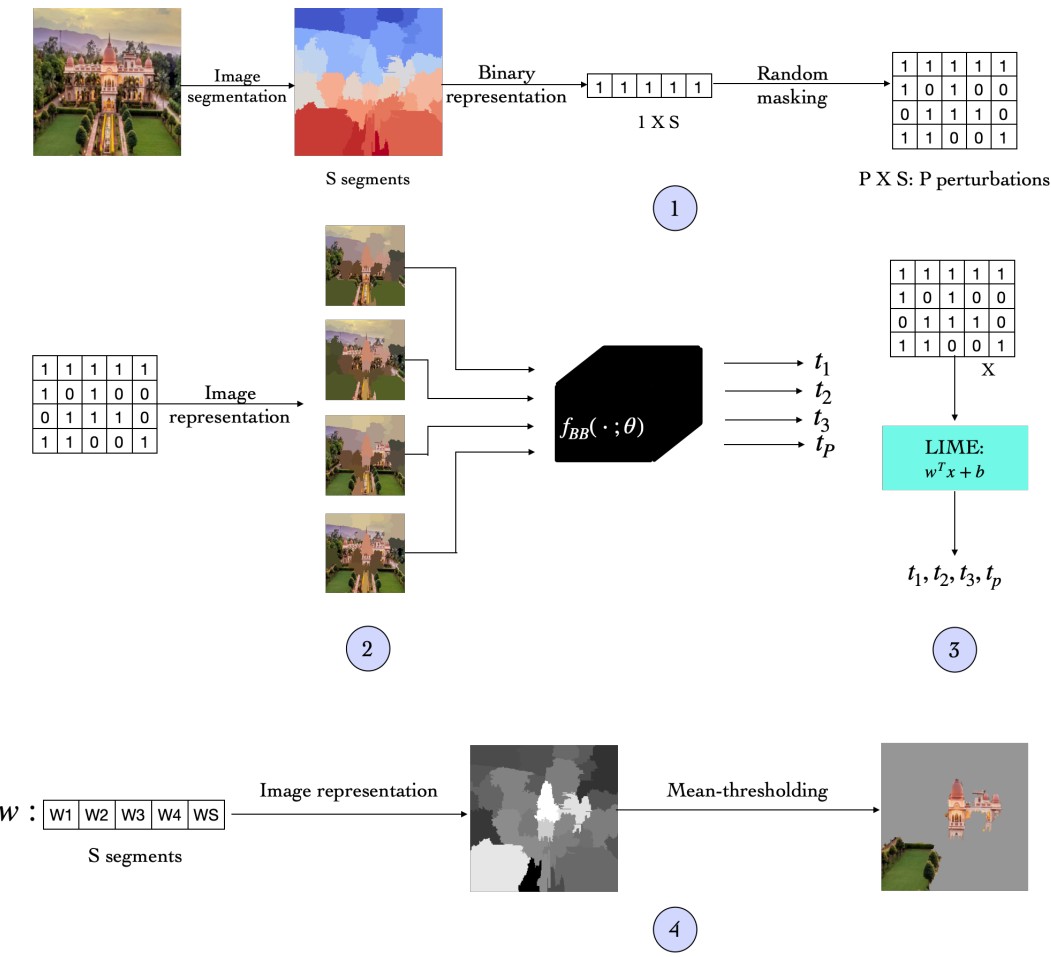

Figure 7: Steps involved in generating explanation using LIME Ribeiro et al. (2016)

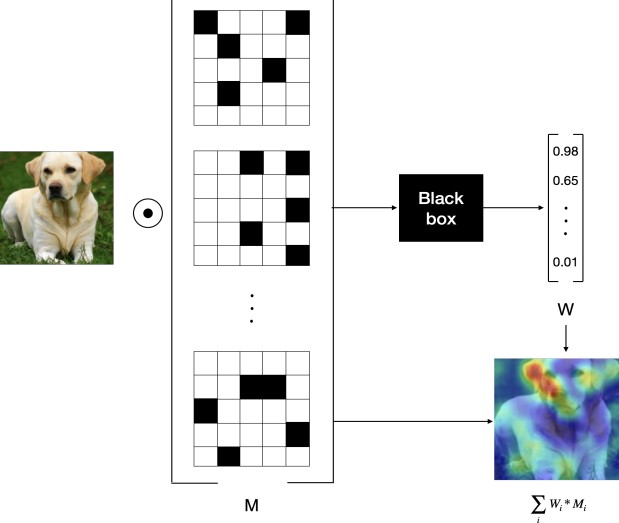

Figure 8: Steps involved in generating explanation using RISE Petsiuk et al. (2018)

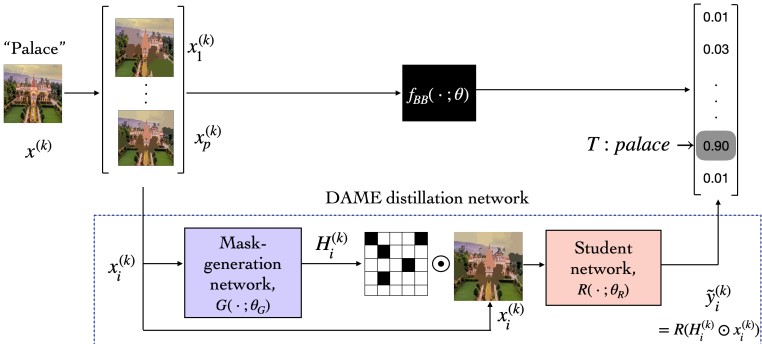

Figure 9: DAME, operating in a teacher-student based distillation framework, finds the saliency explanation using the mask generation network, and the student network distills the black box locally.

- RISE first generates $M$ random masks. Then, the masked images generated using them are fed to the black box and the corresponding target class confidences are stored.
- A mask that leads to more black box confidence is more important. Hence, the explanation generated by RISE is simply $\sum_{i=1}^{M} w_i M_i$ as shown in Figure 8.

## A.4 DAME: *learning* THE EXPLANATION

Based on the learnable explainability framework, the DAME student network generates the explanation following Algorithm 1 stated in main draft. The block diagram showing the working of DAME is shown in Figure 9. The training is performed through iterative optimization. An example loss pattern (for the $strawberry$ image) with iterates is shown in Figure 10. The corresponding progression of the explanation is also shown in Figure 11. We observe that the saliency map converges quickly, within

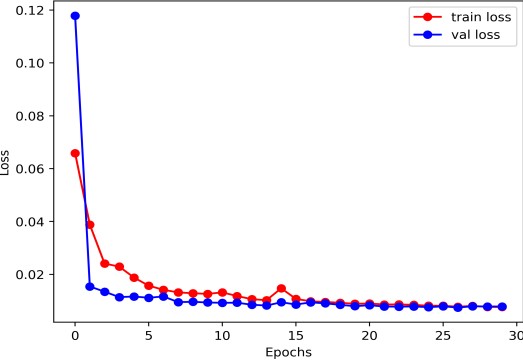

Figure 10: Epoch-wise loss plots during training the DAME student network.

$3 - 4$ epochs. Subsequently, the map becomes sharper and more selective as the epochs progress. For our qualitative and quantitative analysis, we restrict to 10 epochs.

## A.5 HYPER-PARAMETERS

The batch size and number of samples to use for training DAME are the major hyper-parameters. We tune them based on mean IoU performance on a held out validation set of VOC Everingham et al. (2015) segmentation dataset. The corresponding mean IoU results, using ResNet 101 finetuned model as black box, are shown in Tables 4 and 5. It is interesting to note that the gain is prominent when number of neighbourhood samples increase, and it saturates beyond a point. To save computation, 6000 samples are chosen with batch size 32 for our experimentation, although increasing samples further may give slight improvement.

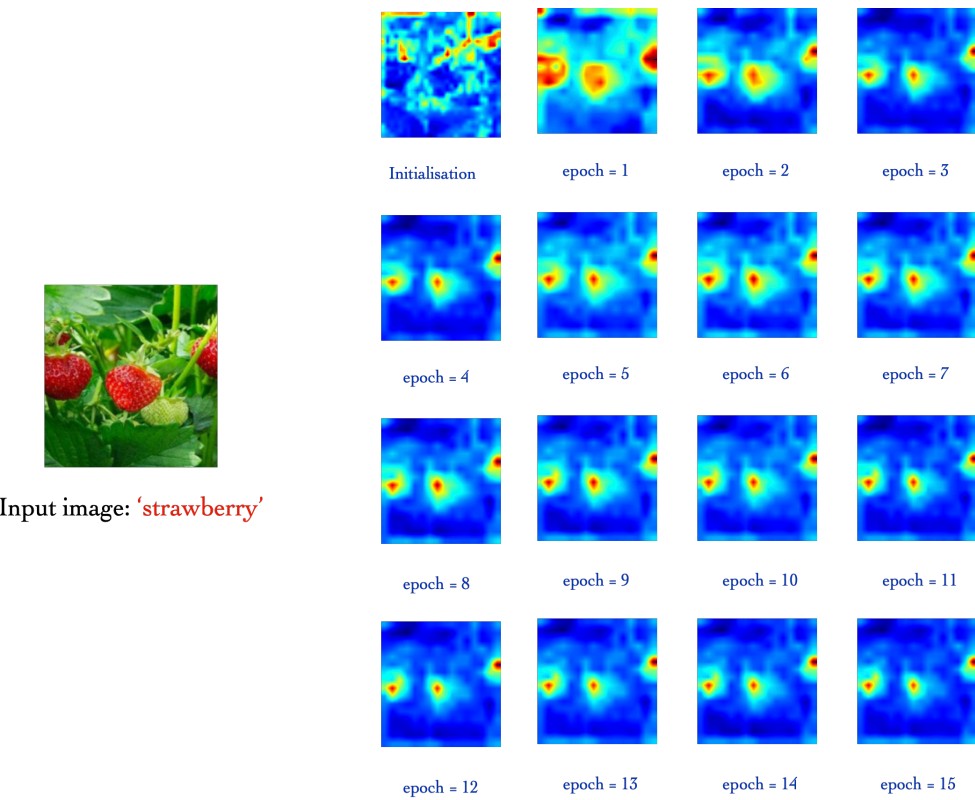

Input image: 'strawberry'

Figure 11: Epoch-wise progression of the saliency explanation for target class 'strawberry'.

Other two hyper-parameters involved are $\lambda$ and $\mu$ in the loss function (equation 7 in main draft). Using similar exercise, they are fixed to $\lambda = 0.001$ and $\mu = 0.02$.

| No. of neighbourhood samples | Batch size | mean IoU |
|---|---|---|
| 500 | 16 | 0.102 |
| 1000 | 16 | 0.132 |
| 2000 | 16 | 0.176 |
| 4000 | 16 | 0.287 |
| 6000 | 16 | 0.304 |
| 10000 | 16 | 0.306 |
| 500 | 32 | 0.091 |
| 1000 | 32 | 0.128 |
| 2000 | 32 | 0.181 |
| 4000 | 32 | 0.293 |
| 6000 | 32 | 0.325 |
| 10000 | 32 | 0.325 |

Table 4: The batch size and number of neighbourhood samples are chosen based on mean IoU performance on the train split of VOC segmentation dataset.

## A.6 EXPERIMENTAL SETUP

### A.6.1 OBJECT CLASSIFICATION TASK

For the qualitative analysis, the ImageNet Deng et al. (2009) classes are used. A pre-trained ResNet-101 model trained on ImageNet is directlty used for this purpose. Given the model and an

| $\lambda$ | $\mu$ | IoU on validation set |
|---|---|---|
| 0.0001 | 0.01 | 31.3 |
| 0.0001 | 0.02 | 32.1 |
| 0.0001 | 0.04 | 32.0 |
| 0.001 | 0.01 | 31.7 |
| 0.001 | 0.02 | **32.5** |
| 0.001 | 0.04 | 32.2 |
| 0.01 | 0.01 | 28.9 |
| 0.01 | 0.02 | 29.6 |
| 0.01 | 0.04 | 30.0 |

Table 5: The hyper-parameters of loss function are chosen based on mean IoU performance on the train split of VOC segmentation dataset.

XAI algorithm, the explanations are generated and compared, as shown in Figure 4 in main draft. Additional results are also provided in Figures 15, 17 and 18 in the Appendix.

The quantitative experiments are based on Pascal VOC 2012 dataset Everingham et al. (2015) as the dataset comes with human annotated ground truth object segmentation masks. The experimental setup involves the following components:

- **Dataset:** The Pascal VOC dataset has 20 classes of objects. The dataset has two components: classification dataset and segmentation dataset. Both the components have train, val and trainval splits. The classification dataset has 11540 images and the segmentation dataset has 2913 images spread across 20 classes. The classwise statistics can be found in voc.

- **Black box classifier:** We use pre-trained classifiers trained on ImageNet and then finetune them on the classification dataset of Pascal VOC dataset. In particular, we use ResNet 101 He et al. (2016) and ViT-16-base Dosovitskiy et al. (2021) models. The final classification layers of the models are modified with 20 output nodes. Post fine-tuning, the ResNet 101 and ViT-16-base achieve $86.1\%$ and $92.3\%$ classification accuracy respectively.

- **IoU based evaluation:** The segmentation dataset is used to evaluate the explanation generated by different methods, including DAME. Given an input image, a black box classifier and an XAI method, the explanation is generated. The black box classifier is chosen among the two fine-tuned classifiers, and the XAI method is chosen among 10 different methods belonging to 3 categories to present an exhaustive benchmark comparison. To allow use of the ground truth segmentation masks as good ground truth explanations, we only consider input images which are correctly predicted by the black box with high prediction confidence. The IoU metric is used for evaluation using the explanations generated and the corresponding ground truth segmentation masks as shown in Figure 14.

### A.6.2 Sound event detection task

The experiments on sound event detection task are based on Environmental Sound Classification (ESC-10) dataset. The experimental setup involves the following components:

- **Dataset:** The ESC-10 dataset has 10 classes of sound events. The dataset has train, val and test splits.

- **Black box classifier:** We use pre-trained classifier trained on ImageNet and then finetune it on the ESC-10 dataset. In particular, we use ResNet 50 model. The final classification layers of the model is modified with 10 output nodes. Post fine-tuning, the ResNet 50 achieves $98.5\%$ classification accuracy.

- **IoU based evaluation:** The ESC dataset for the audio task does not have VOC-like segmentation masks. However they are short duration audio clips with almost full occupancy of the sound events. As single audio clip contains one class of audio event, under the limitation that annotations are unavailable, we assume the audio clip to fully contain the

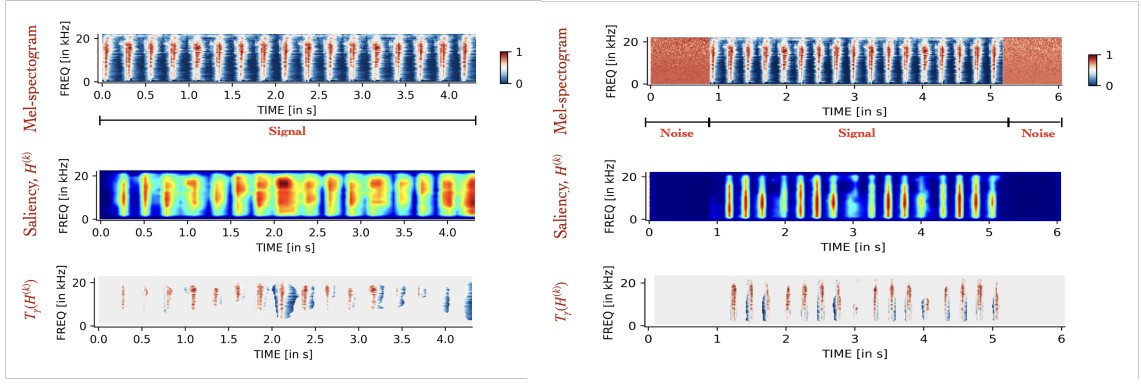

Figure 12: Saliency explanation and it's binary version obtained using DAME for the audio clip with ground truth class soft-max probability of $0.654$. The modified audio changed the black box prediction minimally (soft-max probability from $0.654$ to $0.638$).

audio event. Then, we pad the audio clips from both sides and consider padded time regions as false regions. This way, we simulate the VOC-like experimental setup where the original time region serves as the true region and the padded regions serve as false regions. Following this, the IoU based evaluation is adopted like the vision task. The qualitative analysis of the saliency explanations generated are also shown in Figure 12.

### A.6.3 COVID-19 DETECTION FROM COUGH SOUNDS

The experiments on COVID-19 detection task from cough sounds are based on annotation dataset provided by Zhu et al. (2022). They manually listen to cough sounds of COVID-19 positive and healthy subjects and annotate them on time axis based on sound events (*couughing*, *throat clearing*, *noise* etc.). The experimental setup involves the following components:

- **Dataset:** We use the annotated DiCOVA challenge dataset Zhu et al. (2022); Sharma et al. (2022).

- **Black box classifier:** We use the bidirectional long short-term memory (BLSTM) based classifier provided as part of the challenge. The classifier achieves $79.8\%$ area under the receiver operating characteristics curve (AUC-ROC) for the COVID-19 diagnosis task.

- **IoU based evaluation:** The sound event annotations provide the opportunity to find out regions where the classifier puts more emphasis while making the decision. The classifier is trained on overlapping segments obtained by breaking a audio clip into multiple overlapping segments. As it doesn't take full audio clip as input, using gradient based approaches to generate explanation is not straightforward. However, gradient-free perturbation approaches can be used easily. With the help of provided annotations, we compute the IoU values based on different sound events as reported in main draft.

### A.7 SUBJECTIVE EVALUATION OF EXPLANATIONS

A total of 35 subjects are chosen to evaluate 28 saliency explanations provided by different post-hoc, model agnostic, gradient-free methods as shown in Figure 13. The subjects are provided with the prediction by the black box and the explanations. The DAME explanations are preferred by them over other methods, statistically significantly, as discussed in the main draft.

### A.8 QUANTITATIVE EVALUATION OF EXPLANATIONS USING IoU METRIC

We illustrate the computation of the intersection over union (IoU) metric for comparing the model explanation and the human annotated ground-truth in Figure 14. The IoU metric is a number between $[0, 1]$ with a larger value desired. The value of $1.0$ is achieved only the reference mask fully overlaps with the model predictions.

## Subjective test

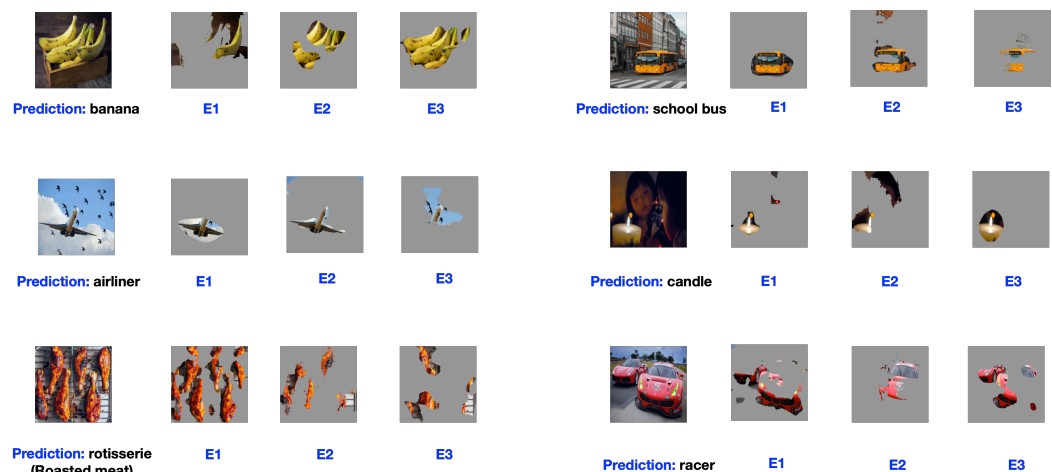

Figure 13: Subjective test format, provided to the subjects during evaluation.

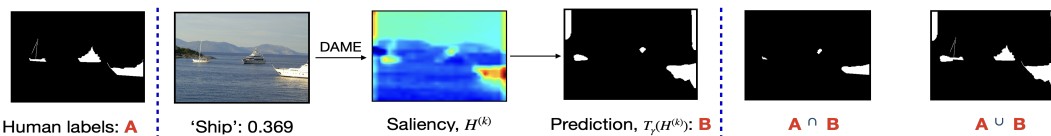

Figure 14: Let's denote $A$ as the ground truth segmentation map for the class "ship". Given a black box, trained on ImageNet dataset, $H^{(k)}$ denotes the saliency obtained using an XAI method (e.g. DAME). By setting a threshold, $\gamma$, over the $H^{(k)}$, we can calculate the predicted segmentation map $B = T_\gamma(H^{(k)})$. Then, the IoU can be computed as, $IoU(A, B) = \frac{A \cap B}{A \cup B} = \frac{4154}{18213} = 0.228$.

### A.9 EXPLANATIONS VISUALISATION IN ORIGINAL FORMS

Figure 4 in main draft comparatively analyzes explanations generated by different XAI methods. Further, extensive comparison with respect to different gradient methods are included and shown in Figure 15. However, the XAI methods under comparison come from different categories (gradient based explanations, saliency, feature importance etc.). Hence, to ease comparison, the part of images using thresholded explanations are shown. The original forms of the explanations are shown in Figure 16 here.

### A.10 ADDITIONAL RESULTS

Qualitative analysis was shown for 16 images in main draft. To allow further analysis and comparison, Figures 15, 17 and 18 show more results on a diverse set of inputs. Following are the key observations,

- As shown in Figure 16, the DAME generated explanations are more precise as compared to most of the other methods.
- In particular, because it operates on the image space, it is much more precise than local approximation techniques like LIME which operates on segment masking information.
- For few input images, it is able to generate the most accurate explanation compared to others, for example, $banana$, $airliner$ images in Figure 17 and $power\_drill$, $rotisserie$ and $tractor$ images in Figure 18.

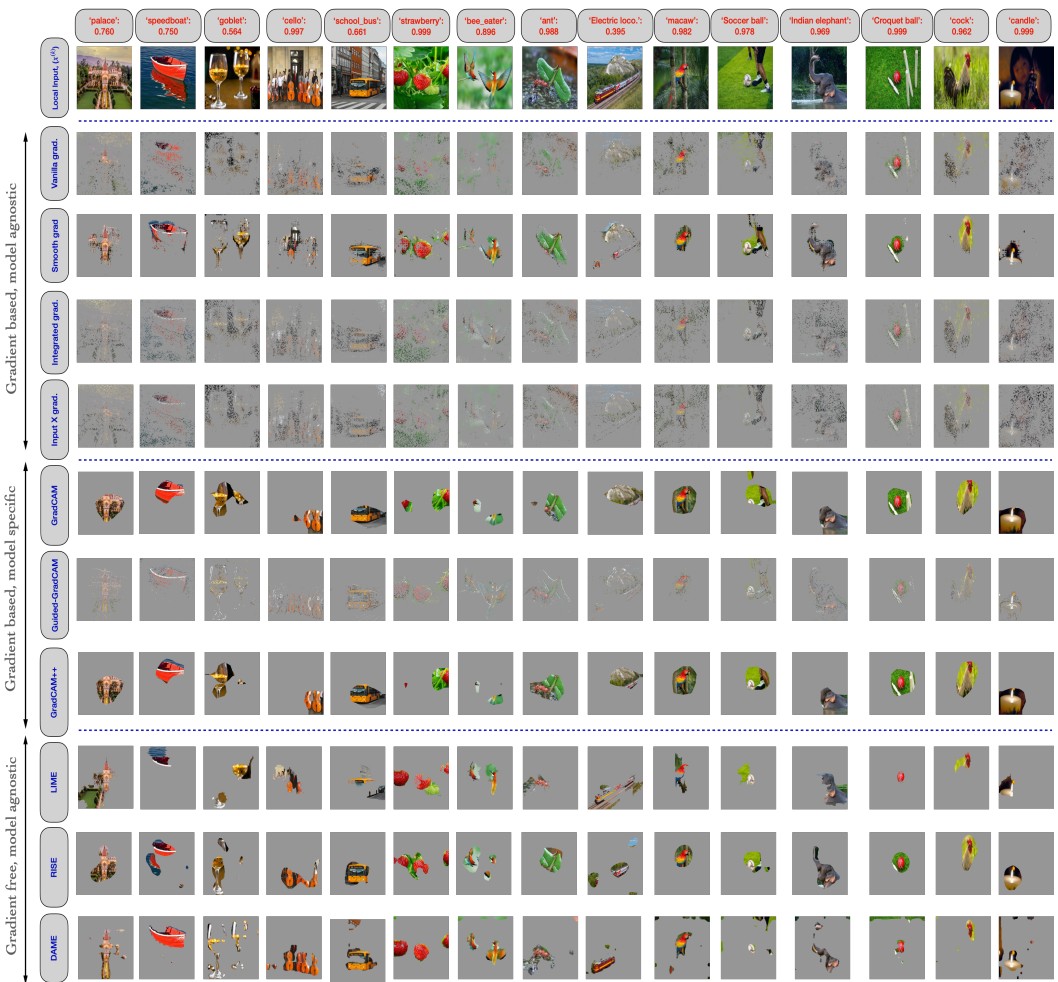

Figure 15: Qualitative assessment of 9 XAI methods from 3 different categories, along with DAME, applied to the ResNet-101 black-box model. The target class for which explanation is sought and the corresponding black box confidence scores are mentioned. The explanations obtained from feature importance based methods (e.g. LIME) and saliency based methods (e.g. GradCAM) are thresholded and imposed on input images. The original explanations are shown in Appendix.

- For images with multiple object instances, DAME explanations are more complete than other methods, for example, $lemon$ and $rotisserie$ images in Figure 18.

## A.11 HARDWARE DETAILS

We used 1 Nvidia $A6000$ and 2 Nvidia $A5000$ GPU cards for our experimentation and ablation studies.

## A.12 DELETION EXPERIMENT

We clarify the evaluation carried in the draft and add further evaluations based on the suggestions given by the reviewer. For the results reported in the main draft, we had removed the most important 30% pixels (as obtained using any of the XAI methods) at once.

**Reason behind** 30% **feature deletion**: We provided this quantitative evaluation on the Pascal VOC dataset. The dataset comes with images ($x^{(i)}$) along with object annotations ($Z^{(i)}$), as shown in

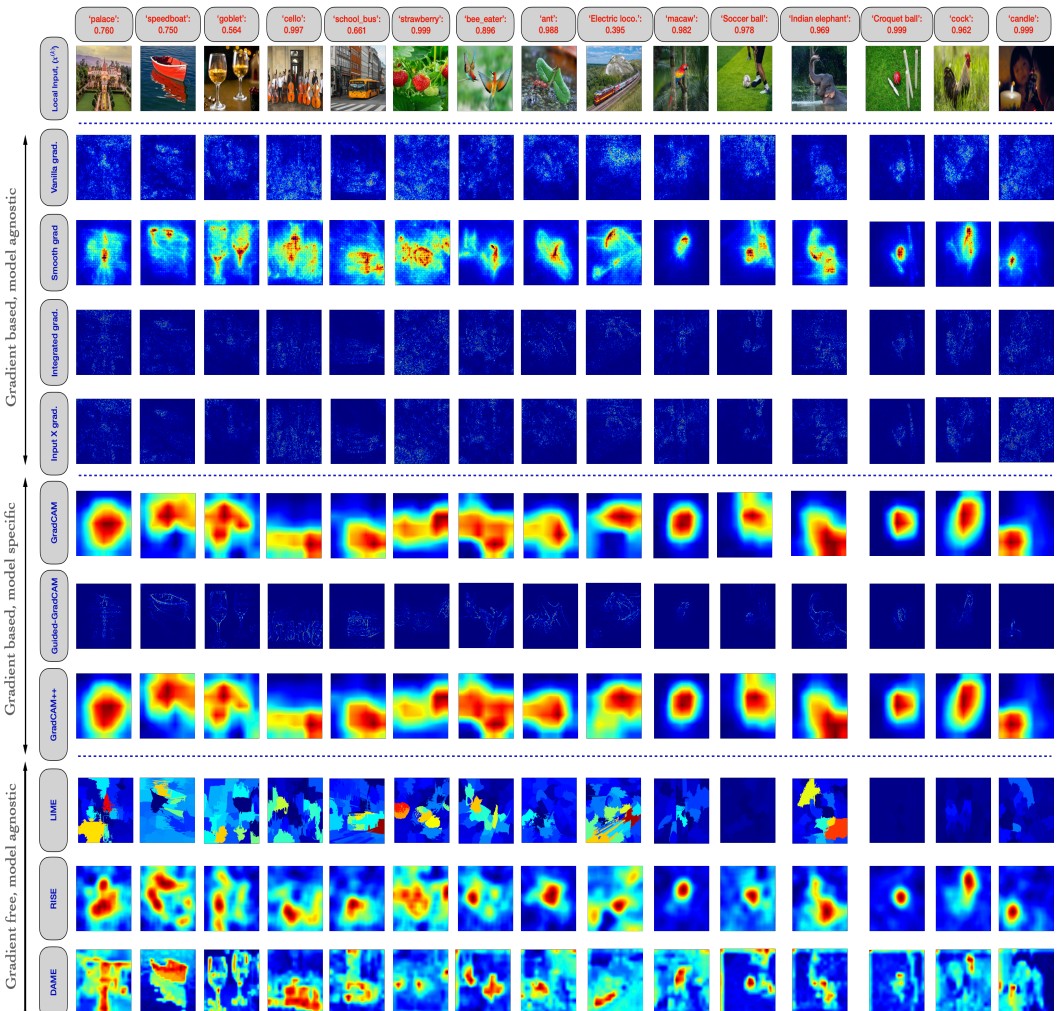

Figure 16: The original forms of the explanations shown in Figure 4 in main draft.

Figure 20. Let the object occupancy in the images, defined as,

$$\mu_O^{(i)} = \frac{|(m,n) : Z^{(i)}(m,n) = 1, (m,n) \in Z^{(i)}|}{|(m,n) : (m,n) \in Z^{(i)}|} X100\% \tag{10}$$

Then, the mean object occupancy $\mu_O = \frac{1}{|D|}\sum_{k=1}^{|D|}\mu_O^{(i)}$ is found to be $29.2\%$ ($D$ denotes the VOC dataset) as shown in Figure 20. Hence, if object occupancy is known, removal of $\mu_O$ ($\sim 30\%$ for VOC) important features provides a fast, one-shot approach to evaluate explanations. For example, if gradual deletion is performed in $x$ number of steps, it will require $x$ number of inferences for every image.

However, as pointed to by the reviewer, we realize that gradual deletion is a standard way of fidelity evaluation. We performed the progressive deletion process as reported in [1], in steps of $10\%$, and found that in terms of AUC, RISE is better than LIME as shown in Figure 19. Petsiuk et al. (2018) reported only drop in model confidence for explanation class along y-axis. However, we have reported drop in accuracy too. Also, we note that plots and AUC values in [1] are for some select examples, whereas we have reported the metrics over the entire VOC dataset. It is seen that among the three methods, the DAME shows the most sensitivity to deletion.

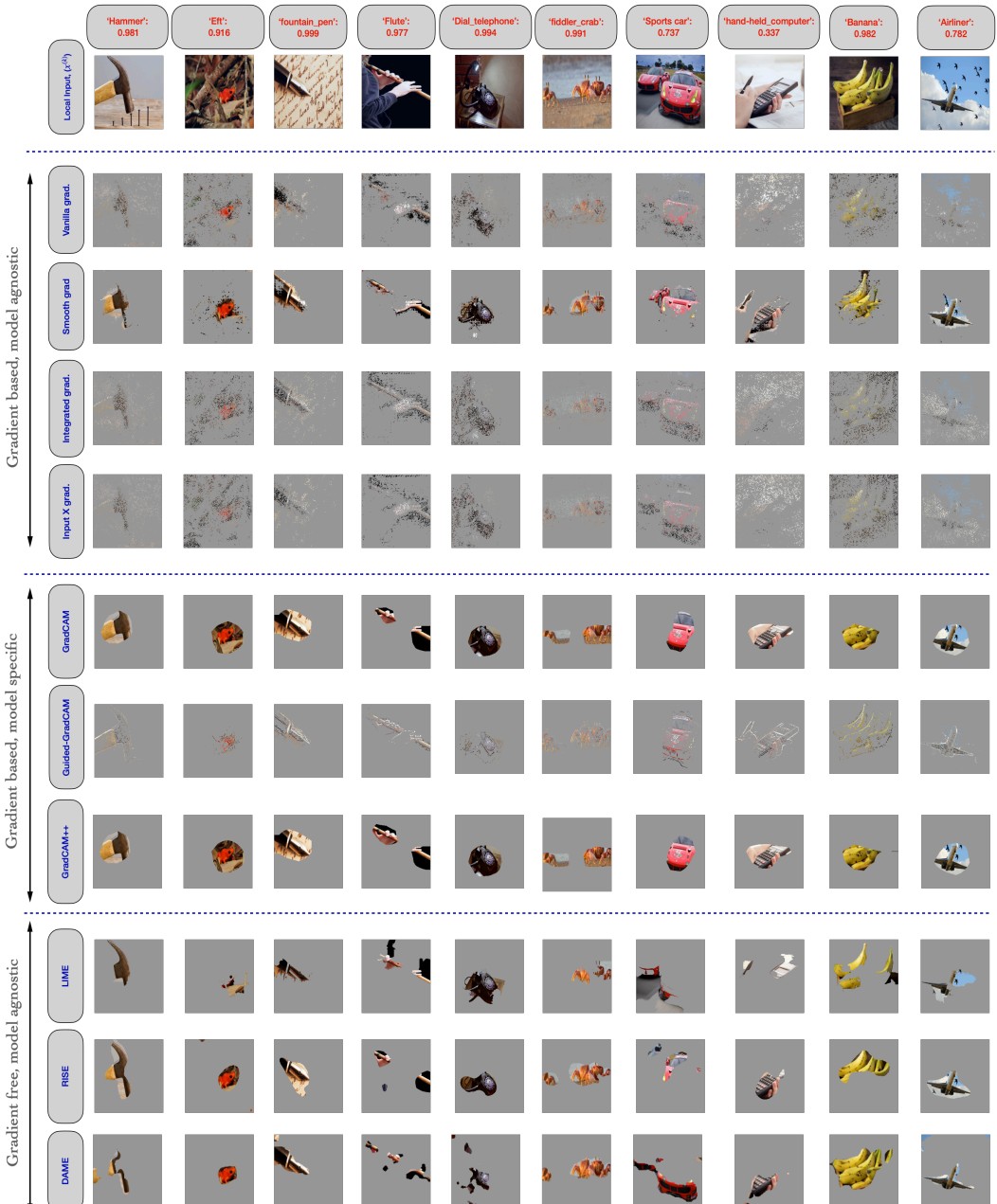

Figure 17: Qualitative assessment of 9 XAI methods from 3 different categories, along with DAME, applied to the ResNet-101 black-box model. The target class for which explanation is sought and the corresponding black box confidence scores are mentioned at top.

## A.13 DAME PERFORMANCE USING DIFFERENT LOSS COMPONENTS

Thank you for the question. Although it is true that the loss function has three components, we found that tuning them for a modality works well for multiple applications with that modality. For example, the hyperparameters tuned for ImageNet was used as is for VOC dataset, and hyperparameters tuned for ESC-10 based audio classification were used as is for the COSWARA application.

Moreover, we would also like to add that the KL-div loss component contributes minimally as can be seen from the Table 6. Hence, the MSE loss along with a L1 loss component (to avoid singularity) itself may constitute a reasonable choice for the loss function in DAME.

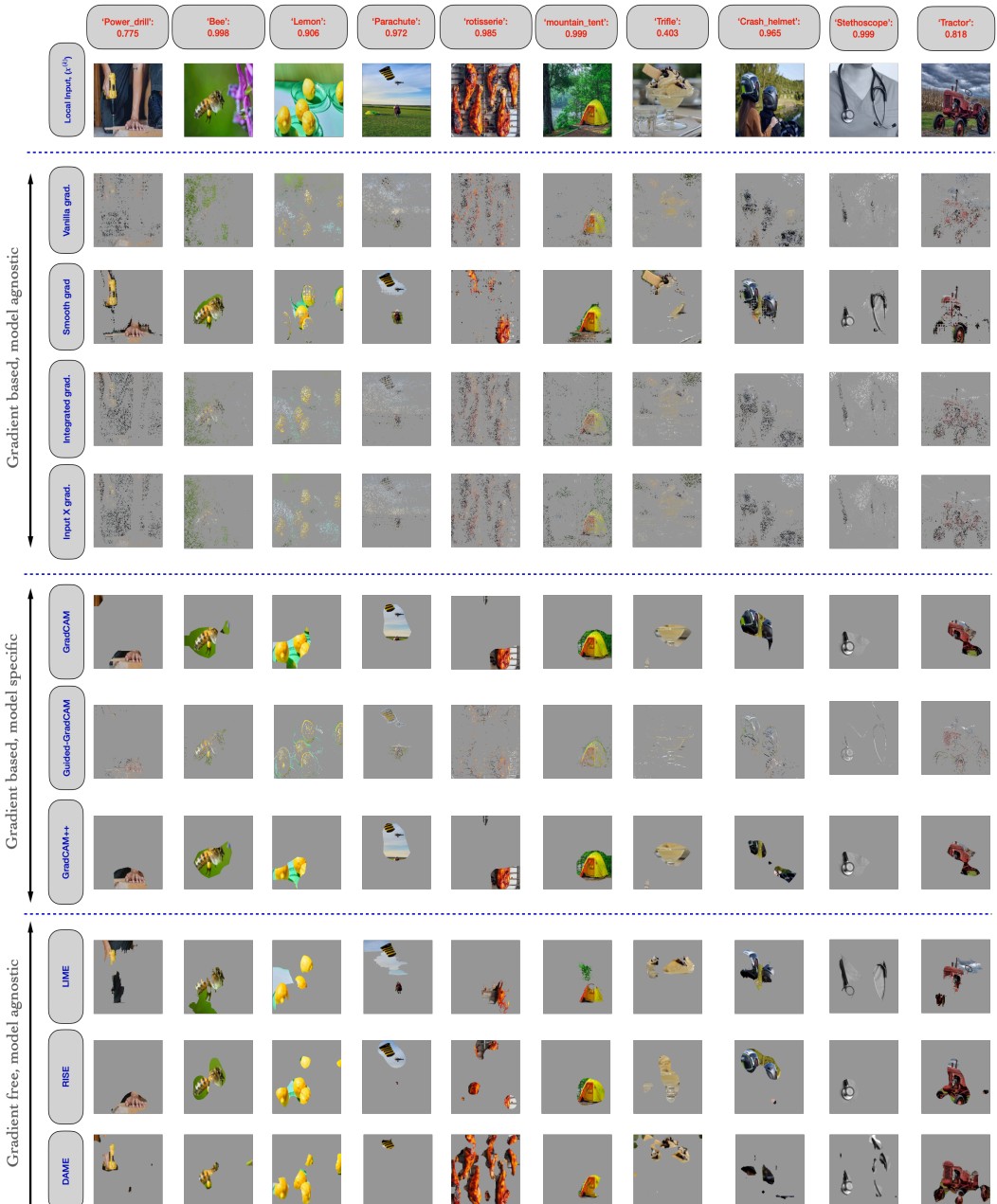

Figure 18: Qualitative assessment of 9 XAI methods from 3 different categories, along with DAME, applied to the ResNet-101 black-box model. The target class for which explanation is sought and the corresponding black box confidence scores are mentioned at top.

| DAME variants | ResNet-101 | ViT base-16 |
|---|---|---|
| Loss: MSE+$\lambda$.L1 | 32.4 | 30.8 |
| Loss: MSE+$\lambda$.L1+$\mu$.KL-div | 33.3 | 31.4 |

Table 6: IoU values obtained on VOC dataset with and without KL-div loss using DAME.

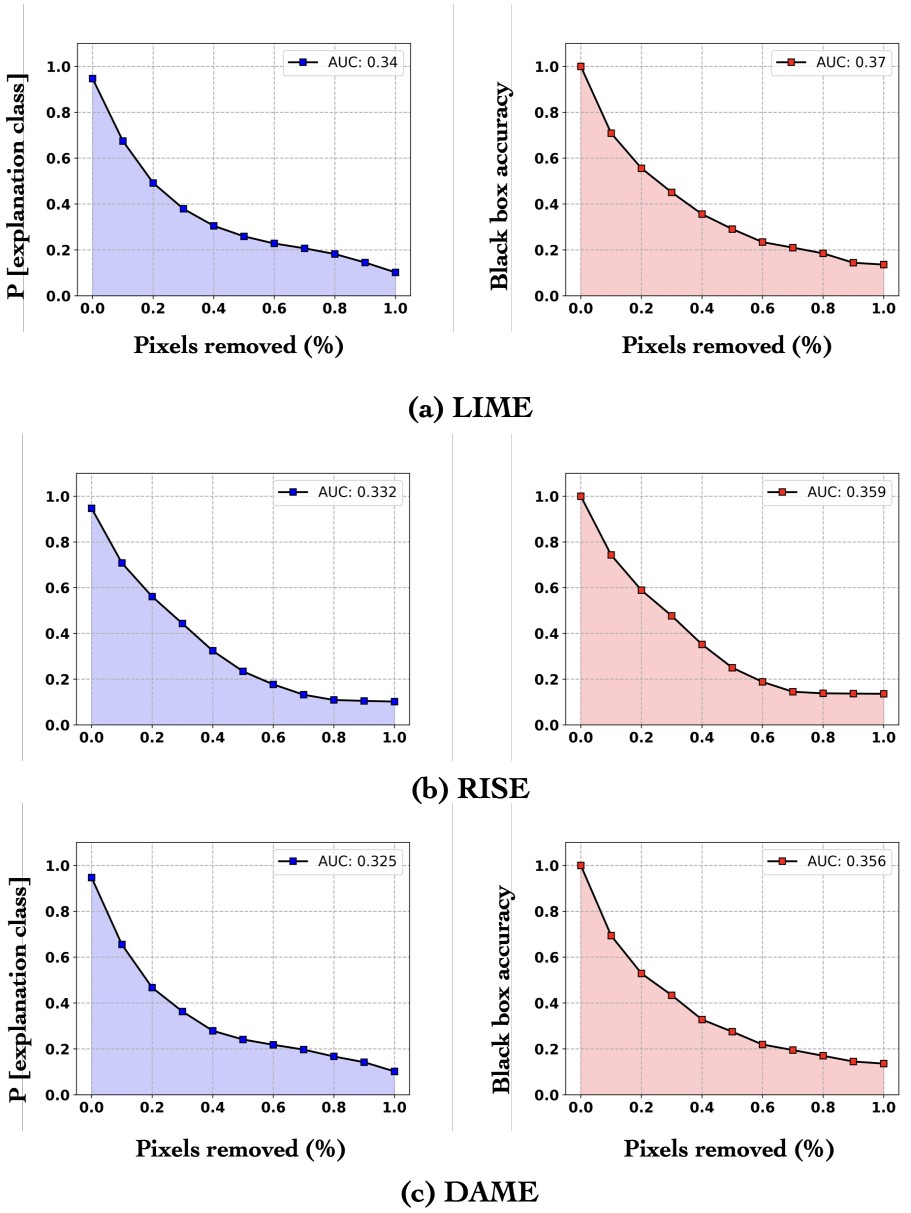

Figure 19: Gradual deletion based measure of XAI methods on VOC dataset. The mean accuracy drop and corresponding mean confidence drops are reported. A lower value of AUC value corresponds to better explanations.

## A.14 DAME FOR CORRELATED FEATURES

Examining sensitivity of XAI methods with respect to correlated features is an important area of research. While a detailed quantitative evaluation was not possible within the short period, we have resorted to a qualitative evaluation. Following is an example in Figure 21, where a 'cat' image along with the shadow associated with it. The shadow has similar shape as the cat, leading to a correlated feature in the input. However, the DAME generated explanation is not influenced by the presence of the shadow.

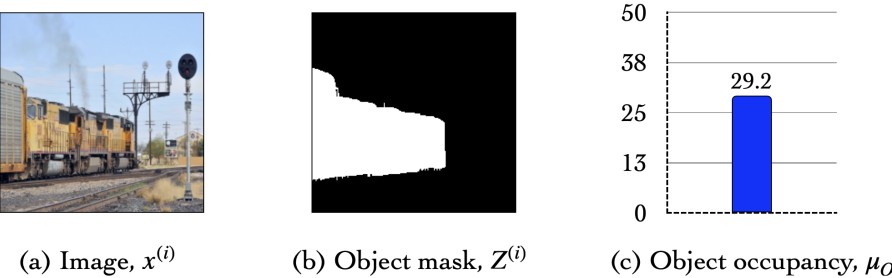

(a) Image, $x^{(i)}$      (b) Object mask, $Z^{(i)}$      (c) Object occupancy, $\mu_O$

Figure 20: The VOC dataset provides (a) image examples ($x^{(i)}$), and (b) object annotations ($Z^{(i)}$). (c) The average object occupancy is found as $29.2\%$.

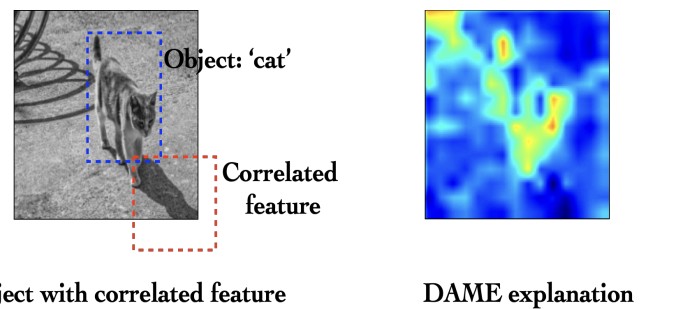

**Object with correlated feature**      **DAME explanation**

Figure 21: DAME generated explanation for an image with correlated feature.

### A.15   SCORE AGREEMENT WITH BLACK BOX

We carried out experiments to examine how much the model based XAI methods (LIME and DAME) agree with the black box behavior for perturbed samples- commonly used as one of the fidelity measures. LIME and DAME tries to mimic the black box at the local neighbourhood and hence their response ideally should match with black box response for samples drawn from local neighbourhood of the input. We observe a substantially stronger agreement for DAME over LIME, as shown in Figure 22.

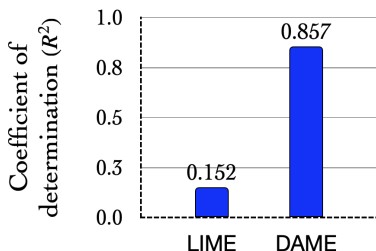

Figure 22: $R^2$ value denoting the agreement between black box confidence and XAI model confidence, on the perturbed samples. DAME shows significantly more agreement than LIME.

