# OpenReview forum: "DAME: A Distillation Based Approach For Model-agnostic Local Explainability"
_ICLR.cc/2024/Conference — Submitted to ICLR 2024_

### Official Review · Reviewer_7xDX · 2023-10-29

**Soundness:** 2 fair
**Presentation:** 3 good
**Contribution:** 2 fair
**Rating:** 5
**Confidence:** 4

**Summary:**

Explaining a deep neural network decision on a data point by using linear models as approximators in the locality of the data point has become a common practice. This paper argues that local linear approximation is inapt as the black boxes under investigation are often highly nonlinear. They propose a novel local attribution methods Distillation Approach for Model-agnostic Explainability (DAME) which does not use a linear model as local approximator. The method consist of training a student network to copy the prediction of the original DNN on the perturbated version of the data point along with a Mask-generator network that masks those perturbated samples. After training, this Mask-generator will be used to generate an explanation for the original DNN. DAME is evaluated on computer vision datasets using (a) IoU between the explanation and human annotation, (b) human subjective rating of the quality of the explanation, and (c) the drop in accuracy of the original DNN when the important pixels are removed. They also evaluate it using an audio dataset and a medical dataset, on which both use an IoU metric.

**Strengths:**

- The paper is clear and the work is well contextualized regarding prior works (although it could refer to more recent perturbation-based attribution methods).
- Using distillation methods for explaining a model is an interesting idea

**Weaknesses:**

- The main weakness of this paper is in the evaluation.
	- It is because we do not know the reasoning behind DNN decisions --i.e. we do not what a good explanation of its decision is-- that we carefully develop methods for that purpose. In that sense, a subjective human evaluation of the quality of the explanation (b) is not actually informative of the quality of the explanation
	- The decisions of a DNN do not necessarily rely on the same features humans rely on (the opposite has previously been shown [1-2]). Hence an explanation that accurately depicts that the DNN does not use human-like features will be wrongly penalized by IoU metrics (a)
	- On the other hand, a standard way to evaluate attribution methods is using fidelity measure, Deletion and Insertion --introduced in RISE-- being the most widely used ones, which the paper does to compare DAME with RISE and LIME (it is not exactly clear if pixels are progressively removed as in Deletion or if all important pixels are removed at once). If Deletion is indeed used, the results of the 2 baseline are slightly surprising as RISE has been shown consistently to be better than LIME in previous work [3-4], which is not the case here.
- Also, the motivation of the paper comes from the claim that linear models are inapt to accurately approximate non-linear models locally. An instantiation of the proposed framework with linear models is missing to make the claim more concrete.


[1] Geirhos et al. Shortcut learning in deep neural networks. Nature Machine Intelligence. 2020.

[2] Fel et al. Harmonizing the object recognition strategies of deep neural networks with humans. NeurIPS. 2022.

[3] Petsiuk et al. RISE: Randomized input sampling for explanation of black-box models. BMVC. 2018

[4] Novello et al. Making Sense of Dependence: Efficient Black-box Explanations Using Dependence Measure. NeurIPS. 2023

**Questions:**

- I was wondering if the authors have thought about running standard attribution methods on the original and student models as a sanity check that they do seem to have similar decisions for similar reasons?

---

> ### Author Response · Authors · 2023-11-19
> **Addressing the questions**
>
> We thank the reviewer 7xDx for reviewing the draft and providing valuable comments.
>
> 1. ***clear, well contextualized***: Thank you for the appreciating our work.
>
> 2. ***subjective evaluation not informative***: Thank you for the pointer on the evaluation.
> * We agree that there is no single way to explain away the decisions made by models.  Hence, we evaluate in a multi-pronged way, using fidelity measure, IoU, and the human subjective study.
> * The motivation for the human evaluation is that, for the samples in which the black-box model is highly confident, how do the humans evaluate saliency maps generated by different XAI methods under the assumption that the black-box model, on these samples with high classification confidence, has attended to the most salient parts of the objects.
> * There are prominent studies and regulations that highlight the importance of explanations being human comprehensible as they are the end users [2,3,4,5].
> * The collective evaluation with multiple metrics highlighted the advantage of DAME.
>
> 5. ***DNN not use human-like features .. penalized by IoU***: The IoU based evaluation is based on the assumption that, the classifier relies on the target object region for condidently right classified samples. To validate:
> * We mask away the regions of the target object annotations, and these masked images leads black box accuracy going down from  $92.3$\% to $18.3$\%.
> * However, masking away the regions outside the object class and keeping the object annotations leads to $76.5$\% accuracy. Although a drop from $92.3$\%, it validates that the annotations as primary information used by ViT model to classify the objects.
> * As pointed by the reviewer, the misclassification may be due to the short-cut learning. To probe this, we computed the IoU on the mis-classified samples while explaining them using ground truth labels and obtained IoU of $16.3$\% ($31.4$\% on correct samples) using DAME. It indicates that the model has failed to focus on the object regions. The drop in IoU for other XAI methods are:
>
> | method | mean IoU | IoU drop from correct samples |
> |------------|----------------|----------------------------------------------|
> | RISE       | 19.7 (14.2)    | 11.4                                         |
> | LIME       | 16.6 (13.2)    | 10.2                                         |
> | DAME       | 16.2 (13.9)    | 15.2                                         |
>
> 6. ***Deletion measure***: We clarify the evaluation carried in the draft are based on removing $30$\% pixels at once.
>
> **Reason behind $30$\% feature deletion**: For images ($x^{(i)}$) and object annotations ($Z^{(i)}$), as shown in Figure 20,
> let the object occupancy is:
> \begin{equation}
> \mu_O^{(i)} = \frac{|(m,n): Z^{(i)}(m, n)=1, (m,n)\in Z^{(i)}|}{|(m,n): (m,n)\in Z^{(i)}|}X 100\%
> \end{equation}
> The mean object occupancy $\mu_O = \frac{1}{|D|}\sum_{k=1}^{|D|}\mu_O^{(i)}$ is found to be $29.2$\% ($D$ denotes the VOC dataset) as shown in Figure 20 and  hence $30$\% important feature removal is used as a fast, one-shot approach. If gradual deletion is performed in $x$ steps, it will require $10x$ inference time.
>
> However, we also did gradual deletion as a standard way and found the (AUC- {**LIME: 0.340, RISE: 0.332, DAME: 0.325**}) as shown in Figure 19. [1] reported only drop in model confidence but we reported drop in accuracy also. Also note the plots and AUC values in [1] are for selected samples, but we have reported the metrics  over full VOC dataset; and DAME shows the most sensitivity to deletion.
>
> 7. ***proposed framework with linear models is missing***: We used a linear student network in DAME and observe that the model fails to generate meaningful explanations. The fidelity measure at $30$\% feature deletion results in only $24.3$\% drop in accuracy as opposed to $56.2$\% for the non-linear model.
> 8. ***similar decisions for similar reasons?***: We removed the top-k features (based on XAI output) gradually from the image and pass the masked images through the DAME student model  as well as the black-box model to get the  scores. The DAME scores show high similarity with the black box scores with $R^2$ of $0.819$ on VOC dataset, confirming their agreement.
>
> References:
> 1. Petsiuk, Rise: Randomized input sampling for explanation of black-box models.
> 2. Nguyen. “The effectiveness of feature attribution methods and its correlation with automatic evaluation scores”. NeurIPS 2021.
> 3. Colin, What i cannot predict, i do not understand: A human-centered evaluation framework for explainability methods. NeurIPS 2022.
> 4. Goodman. “European Union regulations on algorithmic decision-making and a “right to explanation””. AI magazine 2017.
> 5. Doshi. "Towards a rigorous science of interpretable machine learning." arXiv 2017.
> 6. Sun, Ao, "Explain Any Concept: Segment Anything Meets Concept-Based Explanation.", NeurIPS 2023.
> 7. Lerman, Samuel,. "Explaining Local, Global, And Higher-Order Interactions In Deep Learning." ICCV 2021.

---

> ### Comment · Reviewer_7xDX · 2023-11-21
> **Response to author**
>
> Thank you for addressing some of my points, I will raise my score accordingly.
>
> 2. I still stand by my original argument.
> 5. I appreciate the validation done for the IoU measure, but I still do not see its value in evaluating methods when you already use the standard fidelity metrics, especially if a fidelity-like metric is needed to validate it. Again, we are not sure that the model only cares about human-like features and we are not sure that the XAI method highlights the feature used by the model, yet using IoU requires us to make assumptions about either or both of them, hence I would be less confident in the results.
> 6. Thank you for the clarification, the motivation for this choice is sound.
> 7. and 8. Thank you for addressing my questions.

---

> > ### Author Response · Authors · 2023-11-22
> > **Thank you, Reviewer 7xDX**
> >
> > We thank Reviewer 7xDX for his valuable time, going through our response, appreciating, and increasing the score.

---

### Official Review · Reviewer_RyMT · 2023-10-31

**Soundness:** 4 excellent
**Presentation:** 3 good
**Contribution:** 2 fair
**Rating:** 6
**Confidence:** 4

**Summary:**

This paper proposes a model-agnostic, gradient-free, saliency-based method to understand local behavior of black-box models. They establish the shortcomings of previous works like LIME that use a locally linear model to approximate the behavior of a neural network in a given sample’s neighborhood. They use ideas from the MLX (Machine Learning from Explanations) area and propose to address this via distilling the black-box model into a smaller student model only in the sample’s neighborhood. Concretely, they generate perturbations of a sample and then learn saliency masks (explanations) such that a perturbed sample masked by the saliency when passed through the student model has the same target class softmax score as the teacher. These two models, the one that learns the masks and the student that distills the black-box in a sample’s neighborhood, are chained and trained together using a distillation+explanation loss (with 2 more loss terms to avoid identity learning and preserve class distributions between student and teacher). They share results of their approach on 2 vision datasets and 2 audio datasets.

**Strengths:**

1.	This method works in input space and not in the binary mask space like LIME does
2.	The paper establishes the shortcoming of locally linear approximations with a small toy experiment.
3.	They share results from many varied experiments with both quantitative metrics and qualitative samples. To quantify the quality of their explanations, they compute IoU with human annotations for samples that are classified correctly by the model – since that is the class that human annotations would be explaining.
4.	It is an intuitive approach. The paper is well written and easy to follow
5.	They compare with LIME, RISE, GRADCam and other gradient based methods from Integrated Gradients family.
6.	The appendix is very thorough and quite informative

**Weaknesses:**

1.	The biggest bottleneck to using this approach would be having to train a whole new model to understand the behavior of the model for one single input sample.
2.	Results from RISE are often quite competitive in tables 1 and 2. Smooth Grad is also quite competitive.
3.	This approach is akin to a gradient-based approach in the guise of gradient-free. If one was to distill the whole black-box into another model (not just in the sample’s neighborhood) and then apply any gradient-based method, I believe that that would be much simpler since one won’t have to train a smaller model to get an explanation for each sample and I believe it would perform competitively seeing the numbers in tables 1 and 2. So, I have doubts about why the authors have taken this round-about route. It is at least worth it to compare this with works that use distillation to understand models.
4.	I might have missed something here but the audio experiment results don’t seem too convincing:
a.	In task 2, padding noise on two sides is an easy noise pattern to learn/catch.
b.	In task 3, cough data says that it was manually annotated. Are there going to be any plans to release this to enable discussion/reproducibility?
5.	Some language in the paper such as “mildly vs strongly non-linear” is non-standard. This is a small nitpick.

**Questions:**

1.	Have the authors considered using this method on well-known spurious feature detection image datasets like Decoy-MNIST and ISIC?
2.	If one was to distill the whole black-box into another model (not just in the sample’s neighborhood) and then apply any gradient-based method, I believe that that would be much simpler since one won’t have to train a smaller model to get an explanation for each sample and I believe it would perform competitively seeing the numbers in tables 1 and 2. So, I have doubts about why the authors have taken this round-about route. It is at least worth it to compare this with works that use distillation to understand models. I would like to get the authors thoughts on these points.
3.	Can the authors clarify if I have incorrectly interpreted the audio experiments setup or results? Are there going to be any plans to release manually annotated cough data to enable discussion/reproducibility?

---

> ### Author Response · Authors · 2023-11-21
> **Addressing the questions of Reviewer RyMT (1/2)**
>
> We thank the reviewer **RyMT** for reviewing the draft and providing valuable input.
>
> 1. ***works in input space and not in the binary mask space***, ***establishes the shortcoming of locally linear approximations a small toy experiment***, ***results from many varied experiments with both quantitative metrics and qualitative samples***, ***intuitive approach, well written and easy to follow***, ***appendix is very thorough and quite informative***:
>
> Thank you for appreciating our problem formulation, presentation and evaluation on diverse set of tasks and datasets.
>
> 2. ***The biggest bottleneck to using this approach would be having to train a whole new model to understand the behavior of the model for one single input sample.***:
>
> Thank you for pointing this out. Because of unavailability of gradient access to black box in gradient-free explainability, all post-hoc methods (LIME, DAME, and RISE) rely on generating  local perturbations from the  input and approximating the corresponding black box responses. In this setting, DAME takes about $50$ sec. of computation time (single A-6000 NVidia GPU on an Intel server) to train on a single sample. The reference numbers for LIME: 25 sec., and RISE: 40 sec. for the same settings. Hence, DAME is only about $25$\% more expensive compared to baseline method of RISE, while yielding performance gains on various tasks and evaluation metrics listed in the paper and the responses.
>
> 3. ***Results from RISE are often quite competitive in tables 1 and 2. Smooth Grad is also quite competitive.***:
>
> SmoothGrad requires internal access to the model in the form of gradients to generate explanations, which is highly restrictive for various black-box models. The proposed approach is gradient-free post-hoc XAI setting, with only input-output access (as described in the paper, various recent models like GPT are released without any internal access to the model architecture). Hence, gradient based XAI methods merely correspond to upper-bound for gradient free methods. Further, as seen in Table I of the main draft, gradient methods like SmoothGrad perform poorly on Transformer based models like ViT.
>
> With regard to comparison with RISE, the  proposed  DAME framework,
>    * Improves in IoU metric over the RISE on all the audio based evaluations.
>    * Improves over the RISE/LIME on image based evaluations using the fidelity based measures, for example, deletion (please refer to Section A.12 and Figure 19 in revised draft).
>
> Also The XAI for object/event classification should be able to shed some light on mis-classifications by the black box, apart from explaining the correctly predicted samples. To examine this, we explored generating saliency based explanations for the mis-classified samples in VOC dataset and compared them with the ground truth annotations. We found that the mean IoU, using different XAI methods, elicit a drop for all XAI methods, with the largest drop for DAME approach as shown in below Table. It indicates that the black box model may have focused on the contextual regions (outside the target class bounding box) for these samples.
> | method | mean IoU | IoU drop from correct samples |
> |------------|----------------|----------------------------------------------|
> | RISE       | 19.7 (14.2)    |     11.4                                         |
> | LIME       | 16.6 (13.2)    |     10.2                                         |
> | DAME       | 16.2 (13.9)    |     15.2                                         |
> **Table**: IoU for misclassified samples.
>    * Subjective tests indicated improved preference for the explanations provided by DAME over the other methods.
>
> 4. ***This approach is akin to a gradient-based approach in the guise of gradient-free. If one was to distill the whole black-box into another model (not just in the sample’s neighborhood) and then apply any gradient-based method, I believe that that would be much simpler. why the authors have taken this round-about route.***:
>
> Thank you for raising it, which also warrants a discussion in the main draft about the motivation for DAME. We argue why the global distillation is not a feasible choice for post-hoc XAI methods.
>   * Global distillation requires access to the training data of the original teacher (black-box) model.  As DAME is proposed for a post-hoc XAI setting, the requirement of access to the supervised training data might pose a severe limitation to the applicability of the XAI (for example, an XAI for an LLM might need a very training corpus to generate a suitable student model).
>    * Simulating the classification bias and spurious correlation behavior of the larger black-box with a simpler student model globally might be infeasible, which may cause a divergent explanation using the student model compared to the explanations generated for the teacher model.

---

> > ### Author Response · Authors · 2023-11-21
> > **Addressing the questions of Reviewer RyMT (2/2)**
> >
> > 5. ***audio experiment results don’t seem too convincing: a. In task 2, padding noise on two sides is an easy noise pattern to learn/catch***:
> >
> > We argue that noise pattern introduced in the audio (babble noise at $10$ dB SNR) is not significantly different from some of the audio event classes in the ESC-10 dataset. For example, audio event classes like 'rain', 'helicopter', 'chain-saw' and 'sea-waves' had substantial acosutic similarity with the patterns of the noise introduced. Further, as seen by the results in Table 2 of the original draft, the best IoU metric is only $24$\%, indicating that identifying the noise regions out of the audio samples may be a non-trivial task for the XAI models.
> >
> > The other considerations that were used to select this strategy are,
> >    * Adding silence from both sides was not feasible as the XAI methods may simply perform energy based filtering to detect the sound events.
> >    * Adding a different class of sound event on both sides changes the black box prediction of the augmented audio.
> >
> > 6. ***In task 3, cough data says that it was manually annotated. Are there going to be any plans to release this to enable discussion/reproducibility?***:
> >
> > The annotations are already publicly available from the publication "On the importance of different cough phases for COVID-19 detection''. Interspeech 2022 with the public link available in that paper. For the camera ready submission, we will also ensure that all our results are reproducible by providing the source codes for DAME through  open-source platforms.
> >
> > 7. ***Some language in the paper such as “mildly vs strongly non-linear” is non-standard. This is a small nitpick.***:
> >
> > Thanks for the pointer. We will correct them in the final version of the draft.
> >
> > 8. ***Have the authors considered using this method on well-known spurious feature detection image datasets like Decoy-MNIST and ISIC?***:
> >
> > Thank you for raising the interesting question. Studying the effect of bias, spurious correlation in datasets/models on XAI method performance is an interesting area of research.
> > We plan to include this as a future scope of extension of this work.

---

> > > ### Comment · Reviewer_RyMT · 2023-11-21
> > >
> > > Thank you for clarifications and discussions. I have read through this thread and other threads. At this time, I would like to stand by my original rating.

---

> > > > ### Author Response · Authors · 2023-11-22
> > > > **Thank you, Reviewer RyMT**
> > > >
> > > > We thank Reviewer RyMT for his valuable time in going through our responses and appreciating.

---

### Official Review · Reviewer_hRgW · 2023-11-01

**Soundness:** 2 fair
**Presentation:** 3 good
**Contribution:** 2 fair
**Rating:** 8
**Confidence:** 4

**Summary:**

The paper proposes a framework for generating a learnable saliency-based explanations model, which is model-agnostic and requires only black box query access to the model. The framework consists of two models: a mask-generation module that generates the saliency maps and a student network to distill the black-box model's predictions by approximating the black-box model's local behavior near the input sample. The parameters of these two networks are learned by generating perturbations in the neighborhood of a given sample.

**Strengths:**

- The paper addresses an important question on generating saliency map-based explanations with only black-box
access to a model.
- Besides traditional tasks from Computer Vision, the paper also reports results on audio processing tasks.
-  The paper is well-written and easy to follow.

**Weaknesses:**

One of the critical issues with the paper is how they evaluate & the choice of baselines. Even though they consider a
a diverse set of tasks, the authors must add additional experiments to strengthen the paper.

It is hard to see whether the proposed framework offers a clear advantage over the baselines (as explanations are typically subjective).

It would also be essential to understand how architectural changes affect the results.

- Does the architecture of the map generation & student network affect the performance? Does it need to be shallow
or deeper? What are the design considerations for these networks?
- How does the proposed method compare to, say, just distilling a smaller model from the black-box model & then
using the distilled network to generate saliency maps (and use these as explanations for the black-box model?)? This should be a baseline.
- What's the need for a map-generation network in the framework? Can't we distill the black-box model through a
student network exposed to the perturbations?
- The authors should add the above two setups as baselines.

**Questions:**

The proposed framework incurs an additional computation cost but performs worse than a simpler technique like RISE, and the improvement seems marginal.

How important are the perturbations? The mask-generation network seems to be trainable without the perturbations of inputs. It would be better to investigate the impact of the number of perturbations on explanation performance to evaluate the effectiveness of the perturbations.

I also encourage the authors to consider benchmarks like CUB & AwA2 (and other benchmarks where concepts are annotated), which contain annotations of salient parts of the image; this helps them compare against some gold standards

Refer to Weaknesses for additional questions.

---

> ### Author Response · Authors · 2023-11-21
> **Addressing the questions of Reviewer hRgW (1/2)**
>
> We thank the reviewer **hRgW** for reviewing the draft and providing valuable input.
> 1. ***addresses an important question on generating saliency map-based explanations with only black-box access***:
>
> Thank you for appreciating the motivation of our work.
>
> 2. ***Besides traditional tasks from Computer Vision, the paper also reports results on audio processing tasks.***:
>
> Thank you for appreciating our experimental setup on diverse tasks.
>
> 3. ***The paper is well-written and easy to follow.***:
>
> Thank you.
>
> 4. ***Even though they consider a diverse set of tasks, the authors must add additional experiments to strengthen the paper. It is hard to see whether the proposed framework offers a clear advantage over the baselines (as explanations are typically subjective).***:
>
> We have performed a set of additional experiments to further highlight the advantages of DAME over other methods, and to experimentally validate the choices made in the paper.
>    * **Score agreement with black box**: We carried out experiments to examine how much the model based XAI methods (LIME and DAME) agree with the black box behavior for perturbed samples- commonly used as one of the fidelity measures. LIME and DAME tries to mimic the black box at the local neighbourhood and hence their response ideally should match with black box response for samples drawn from local neighbourhood of the input. We observe a substantially stronger agreement for DAME over LIME, as shown in Figure 22 in the revised draft.
>    * **Gradual deletion based fidelity evaluation**: We performed gradual deletion experiment (as proposed in [1]) instead of removing $30$\% of the top features at once (as reported in the original draft). We observed a lower value of area-under-the-curve (AUC) for DAME, demonstrating its improved sensitivity to deletion, as shown in Figure 19 in the revised draft.
>    * **Probing wrongly classified samples**: The XAI for object/event classification should be able to shed some light on mis-classifications by the black box, apart from explaining the correctly predicted samples. To examine this, we explored generating saliency based explanations for the mis-classified samples in VOC dataset and compared them with the ground truth annotations. We found that the mean IoU, using different XAI methods, elicit a drop for all XAI methods, with the largest drop for DAME approach, as shown in Table 2.1 below. It indicates that the black box model may have focused on the contextual regions (outside the target class bounding box) for these samples, leading to mis-classification.
> | method | mean IoU | IoU drop from correct samples |
> |------------|----------------|----------------------------------------------|
> | RISE       | 19.7 (14.2)    |     11.4                                         |
> | LIME       | 16.6 (13.2)    |     10.2                                         |
> | DAME       | 16.2 (13.9)    |     15.2                                         |
> **Table 2.1**: IoU for misclassified samples.
>    * **Justification of map generation network**: Additional experiments regarding the need of the map generation network is performed, which justifies the two network strategy in the DAME framework as discussed in Table 2.3 below.
>    * **Additional ablation experiments**: Architecture choice of student network, and the minimum number of perturbation samples needed for the DAME framework are experimentally established as discussed in Tables 2.2 and 2.4 below.
>
> 5. ***Does the architecture of the map generation & student network affect the performance? Does it need to be shallow or deeper? What are the design considerations for these networks?***:
>
> We experimented with different architectures of the student network. Using a two layer fully connected network (FCN) following a 2 layer convolutional network (CNN) as student network gives significantly better performance than a other architecture chices as shown in Table 2.2 below.
> | **DAME variants**                               | **ResNet-101** | **Vit base-16** |
> |-------------------------------------------------|----------------|-----------------|
> | student network: 2 CNN + 1 FCN layers           | 31.8           | 30.3            |
> | student network: 2 FCN layers                   | 32.2           | 30.9            |
> | student network (paper) : 2 CNN + 2 FCN layers  | 33.3           | 31.4            |
> **Table 2.2:** IoU values obtained using different student networks with DAME.
>
> Regarding depth, a student network which is  deeper, increases the time overhead to generate the explanations for each input sample. As we attempt a local approximation only, a shallow network may be sufficient which also provides significant savings in computation time.
>
>
> **References**:
> 1. Petsiuk et al. RISE: Randomized input sampling for explanation of black-box models. BMVC. 2018

---

> > ### Author Response · Authors · 2023-11-21
> > **Addressing the questions of Reviewer hRgW (2/2)**
> >
> > 6. ***Compare proposed method to just distilling a smaller model from black-box & using the distilled network to generate saliency***:
> >
> > Thank you for raising it, which also warrants a discussion in the main draft about the motivation for DAME. We argue why the global distillation is not a feasible choice for post-hoc XAI methods.
> >   * Global distillation requires access to the training data of the original teacher (black-box) model.  As DAME is proposed for a post-hoc XAI setting, the requirement of access to the supervised training data might pose a severe limitation to the applicability of the XAI (for example, an XAI for an LLM might need a very training corpus to generate a suitable student model).
> >    * Simulating the classification bias and spurious correlation behavior of the larger black-box with a simpler student model globally might be infeasible, which may cause a divergent explanation using the student model compared to the explanations generated for the teacher model.
> >
> > Using a local student model for the sample under consideration and its perturbations is experimented and reported in Table 2.3 below.
> >
> > 7. ***need for map-generation network? Can't distill the black-box model through a student network exposed to perturbations?***:
> >
> > Thank you for pointing this out. We used the student model alone (without the map generation model) to distill directly on perturbations. This locally trained model is then used to generate explanations using LIME/RISE. They prove to be inferior to the ones proposed in the paper (DAME). Also the key observation is, performing explanation and distillation jointly, as performed in the DAME, is superior to the two steps performed in isolation (distillation followed by explanation). The results are in Table 2.3 below.
> >
> > | **Baselines**                       | ResNet-101 | Vit base-16 |
> > |-------------------------------------|------------|-------------|
> > | Baseline 1: student network + RISE  | 31.2       | 30.5        |
> > | Baseline 2: student network + LIME  | 28.5       | 25.9        |
> > | mask-generation + student (DAME)    | 33.3       | 31.4        |
> > **Table 2.3**: Student model with and without mask generation network.
> >
> > 8. ***Add above two setups as baselines.***:
> >
> > Yes, we will add them in the final draft.
> >
> > 9. ***proposed framework incurs additional computation but performs worse than a simpler technique like RISE***:
> >
> > We respectfully disagree with this comment. The proposed framework,
> >    * Improves in IoU over the RISE/LIME on all the audio based evaluations.
> >    * Improves over the RISE/LIME on image based evaluations using the fidelity based measures as well as on explanations for mis-classified samples.
> >    * Subjective tests indicated improved preference for the explanations provided by DAME over the other methods.
> >
> > While the computation cost is indeed higher ($25$\% more), we highlight the potential improvements in XAI quality may justify this. Further, improving the perturbation strategy might potentially allow improvements in the student model training as well as the map generation quality, which form part of our future scope.
> >
> > 10. ***How important are perturbations? The mask-generation network seems trainable without the perturbations. investigate the impact of the number of perturbations on explanation performance.***
> >
> > As the DAME approach requires a map generation model and the student network to be trained locally, this cannot be performed without perturbations.
> >
> > We performed experiments using different number of perturbations. Table 2.4 reports the mean IoU for them. It is observed that about $6000$ local perturbations  (same order as how many other perturbation approaches use) are sufficient to learn the student model effectively.
> >
> > | **No. of neighbourhood samples** | **Batch size** | **mean IoU (%)** |
> > |----------------------------------|----------------|------------------|
> > | 500                              | 16             | 10.2             |
> > | 1000                             | 16             | 13.2             |
> > | 2000                             | 16             | 17.6             |
> > | 4000                             | 16             | 28.7             |
> > | 6000                             | 16             | 30.4             |
> > | 10000                            | 16             | 30.6             |
> > | 500                              | 32             | 9.1              |
> > | 1000                             | 32             | 12.8             |
> > | 2000                             | 32             | 18.1             |
> > | 4000                             | 32             | 29.3             |
> > | 6000                             | 32             | 32.5             |
> > | 10000                            | 32             | 32.5             |
> > **Table 2.4**: Effect of number of neighbourhood samples.
> >
> > 11. ***consider benchmarks like CUB & AwA2***: Thank you for pointing us to these datasets. We are experimenting with them, and will include the results in the final draft.

---

> > > ### Comment · Reviewer_hRgW · 2023-11-22
> > >
> > > Thanks for the detailed response, I would encourage the authors to add these to the draft if possible, I am updating my score.

---

> ### Author Response · Authors · 2023-11-22
> **Thank you, Reviewer hRgW**
>
> We thank Reviewer hRgW for his valuable time in going through our response and appreciating, and also increasing the score. We will surely include the responses in the final version of the draft.

---

### Official Review · Reviewer_kHxm · 2023-11-06

**Soundness:** 3 good
**Presentation:** 3 good
**Contribution:** 3 good
**Rating:** 6
**Confidence:** 3

**Summary:**

This work proposes Distillation Approach for Model-agnostic Explainability (DAME), an approach which fits a non-linear model is fit in the vicinity of an input sample to to explained. The model is fit to obtain a saliency map explanation based on a teacher-student distillation approach which uses a combination of 3 loss functions. The proposed method is comprehensively evaluated on image and audio datasets using a number of evaluation techniques and shows improvement over existing local explainability methods.

**Strengths:**

Overall when the decision boundary is wiggly and inputs are high dimentsional, sparse linear models may not mimic the source model's behaviour around a sample and it makes sense to use a non-linear approach which might provide a better approximation. The approach proposed to generate the local saliency explanation is novel. The evaluation is quite comprehensive including fidelity-based, subjective and qualitative evaluations and comparison with 9 XAI methods.

**Weaknesses:**

Based on the 3 loss functions that need to be handled, it seems likely that the method may not work out of the box (like LIME) and users will probably need to customize/tune hyper-parameters etc. to get the explanations right.

**Questions:**

- How is local vicinity and distance between the given sample and perturbations defined in DAME - is this same as LIME?
- In case of DAME, can the authors comment on local invariance of explanations (do similar inputs yield similar explanations)?
- Would DAME be impacted by correlated featured?
- Instead of using a masking approach to generate perturbations (e.g. LIME), if we have a realistic distribution of perturbed images (e.g. MeLIME), can the DAME pipeline still be used?

---

> ### Author Response · Authors · 2023-11-21
> **Addressing the questions of Reviewer kHxm**
>
> We thank the reviewer **kHxm** for reviewing the draft and providing valuable inputs.
>
> 1. ***The proposed method is comprehensively evaluated on image and audio datasets using a number of evaluation techniques and shows improvement over existing local explainability methods.***: Thank you for appreciating our work and the evaluations performed on on diverse datasets and tasks.
>
> 2. ***it makes sense to use a non-linear approach***, ***The approach proposed to generate the local saliency explanation is novel***, ***The evaluation is quite comprehensive including fidelity-based, subjective and qualitative evaluations and comparison with 9 XAI methods***: Thank you appreciating our approach, novelty and evaluations.
>
> 3. ***Based on the 3 loss functions that need to be handled, it seems likely that the method may not work out of the box***:
> Thank you for the question. Although it is true that the loss function has three components, we found that tuning them for a modality works well for multiple applications with that modality. For example, the hyperparameters tuned for ImageNet was used as is for VOC dataset, and hyperparameters tuned for ESC-10 based audio classification were used as is for the COSWARA application.
>
> Moreover, we would also like to add that the KL-div loss component contributes minimally as can be seen from the Table below. Hence, the MSE loss along with a L1 loss component (to avoid singularity) itself may constitute a  reasonable choice for the loss function in DAME.
> | **DAME variants**    | **ResNet-101** | **Vit base-16** |
> |----------------------|----------------|-----------------|
> | loss: MSE+L1         | 32.4           | 30.8            |
> | loss: MSE+L1+KL-div. | 33.3           | 31.4            |
>
> 4. ***How is local vicinity and distance between the given sample and perturbations in DAME***: We used the same measure of distance between a sample and its perturbations, the L2 distance between them **in the input space**. On the contrary, LIME uses the distance as the L2 distance between them **in the binary space** ($M_{x^{(k)}}$ space, as described in Section 4.2 in the draft).
> The distance measure in input space results in higher sensitivity because masking-off larger regions and smaller segments may not be similar.
>
> 5. ***In case of DAME, can the authors comment on local invariance of explanations (do similar inputs yield similar explanations)?***:
> Yes, we have verified this experimentally that similar inputs indeed generate similar explanations.
> We have also performed an attribution experiment between the student and teacher model.
> We removed the top-k features (based on XAI output) gradually (k increasing from 0 to 100\%) from the image and pass the masked images through the DAME student model  as well as the blackbox model to get the  scores. The DAME scores show high similarity with the black box scores. An average coefficient of determination ($R^2$) of $0.819$ was obtained for the samples in VOC dataset, confirming the agreement between DAME student and the black box model.
>
> 6. ***Would DAME be impacted by correlated featured?***:
> Examining sensitivity of XAI methods with respect to correlated features is an important area of research. While a detailed quantitative evaluation was not possible within the short rebuttal period, we have resorted to a qualitative evaluation. An example image of a cat with correlated features (the shadow associated with it) and the explanation generated by DAME is shown in Figure 21 of the updated draft.
> The shadow has similar shape as the cat, leading to a correlated feature in the input. However, the DAME generated explanation is not influenced by the presence of the shadow.
>
> 7. ***Instead of using a masking approach to generate perturbations (e.g. LIME), if we have a realistic distribution of perturbed images (e.g. MeLIME), can the DAME pipeline still be used?***:
> We thank the reviewer for pointing us to this resource.
> The proposed approach is indeed agnostic to the choice of masking or perturbation generation approach.
> MeLIME proposes an effective perturbation strategy in general, that improves the explanations provided by different methods like LIME. Because this perturbation strategy essentially focuses on generating samples that better capture the representation of local vicinity of the input, it constitutes a promising experiment to explore before the camera ready version of the submission.

---

> > ### Comment · Reviewer_kHxm · 2023-11-22
> >
> > Thanks to the authors for their detailed response and clearly addressing issues raised by all reviewers. I appreciate it. I will continue to retain my score.

---

> > > ### Author Response · Authors · 2023-11-22
> > > **Thank you, Reviewer kHxm**
> > >
> > > We thank Reviewer kHxm for his valuable time in going through our responses and appreciating them.

---

### Author Response · Authors · 2023-11-22
**Common response**

We thank all the Reviewers for spending their precious time in going through our draft, and providing valuable feedback. We tried our best in addressing their concerns.

We summarise below all the suggestions and our responses.

   * ***Problem statement, diverse tasks and evaluations, good presentation, novelty***: The reviewers have appreciated the problem statement of post-hoc, gradient-free local XAI that is persued. A diverse set of tasks (from vision, audio and biomedical domains) and evaluation approaches that are reported in the paper are also appreciated. They have mentioned that motivation behind the need for a non-linear local approximation was clear. The approach we have taken to address that was mentioned as novel by them as well.
   * ***Clarification about hyper-parameter choices, architecture etc.***: A few questions are asked for additional experiments to understand the choice of architectural details and hyper-parameters. We have provided the details with due expeimental results.
   * ***Additional baselines***: They asked to examine different baseline choices around DAME to understand the importance of our approach with respect to them. We have clarified and provided experimental results on them as much we could do within the short time frame.
   * ***Evaluation approaches***: A few clarifications and experimets are sought reagrding the evaluation metrics and approaches used in the paper. We have tried to clarify them and provided additional experiments on this.

Finally, we thank them again for these inputs which significantly helped us in making the work better.

Thank you,

The authors

---

### Meta-Review · Area_Chair_7qNn · 2023-12-14

**Metareview:**

LIME and RISE are black-box local feature attribution that proposes to approximate a deep image classifier $f$ locally on a single point by generating many perturbation samples. This work proposes DAME, a non-linear approximation of $f$ also locally on a single image. The idea is that DAME would fit the original model better than a single linear explanation model often used in LIME.

The paper demonstrates DAME on both images and audio domains and performs both quantitative and qualitative human studies to assess DAME heatmaps. A major disadvantage of DAME, according to the authors (and raised by reviewer `RyMT`), is that it's 25% slower than the competitors due to its training process of deep neural network.

Most reviewers are positive about the contribution of the paper. Reviewer `RyMT ` is particularly concerned with demonstrating heatmaps in the audio domain. I agree that the utility of the interpretability method is unclear in this domain and not evaluated by this paper. Some reviewers (`7xDX`) are concerned that the human evaluation in the image domain is not meaningful because there is no specific task that the humans are requested to do (e.g. in [1][2] humans are required to use feature attribution maps to make classification decisions). The human evaluation asks humans to rate the `quality`, which is a subjective measure instead of an objective, downstream task.

I agree with reviewer `7xDX `. Given that the central claim of the paper is to trade off the a **slower** inference time for a **better** quality heatmap, the evaluation of the heatmaps is the key in this work. However, this point is not well demonstrated in the human study, which is the groundtruth for evaluation.

While being interesting, in its current form, the work needs further evaluation for this claim to be convincing.
Therefore, I recommend `reject`.

----

[1] “The effectiveness of feature attribution methods and its correlation with automatic evaluation scores”. NeurIPS 2021.

[2] HIVE: Evaluating the Human Interpretability of Visual Explanations (ECCV 2022)

**Justification For Why Not Higher Score:**

The claim that feature attribution maps are better is not well supported by the human study.

**Justification For Why Not Lower Score:**

N/A

---

### Decision · Program_Chairs · 2024-01-16

Reject